# Esophagectomy Versus Endoscopic Resection with Adjuvant Therapy for T1b/T2 Esophageal Cancer: A Systematic Review and Meta-Analysis

**DOI:** 10.3390/cancers17040680

**Published:** 2025-02-17

**Authors:** Eagan J. Peters, Madeline Robinson, Noopur Patel, Biniam Kidane

**Affiliations:** 1Department of Medicine, Temerty Faculty of Medicine, University of Toronto, Toronto, ON M5S 3H2, Canada; eagan.peters@mail.utoronto.ca; 2Department of Internal Medicine, Rady Faculty of Health Sciences, University of Manitoba, Winnipeg, MB R3A 1R9, Canada; robin128@myumanitoba.ca; 3Michigan State University College of Osteopathic Medicine, Michigan State University, East Lansing, MI 48824, USA; patelnoo@msu.edu; 4Section of Thoracic Surgery, Department of Surgery, Rady Faculty of Health Sciences, University of Manitoba, Winnipeg, MB R3A 1R9, Canada; 5CancerCare Manitoba Research Institute, University of Manitoba, Winnipeg, MB R3E 0V9, Canada; 6Department of Physiology & Pathophysiology, University of Manitoba, Winnipeg, MB R3E 0J9, Canada; 7Department of Biomedical Engineering, University of Manitoba, Winnipeg, MB R3T 5V6, Canada

**Keywords:** adjuvant therapy, chemoradiotherapy, chemotherapy, endoscopic mucosal resection, endoscopic submucosal dissection, esophageal cancer, esophagectomy, radiotherapy

## Abstract

Surgical removal of the esophagus is the most recommended treatment for patients with early-stage T1b/T2 esophagus cancer. This surgery involves cutting out the esophagus and a part of the stomach to make a new esophagus. This has a high risk for harmful complications. The aim of this study was to assess if partial removal with endoscopy followed by additional therapy (like chemotherapy/radiation) could be a suitable alternative to surgery. We devised a search protocol and review process according to standardized guidelines to find studies of patients undergoing endoscopic removal followed by additional therapy. Information about survival and complications from these studies was collected and analyzed. The combined information showed that 5-year survival was similar between the two treatments. Patients undergoing endoscopic resection with additional therapy experienced fewer complications. Therefore, endoscopic removal followed by additional therapy could be a suitable alternative to surgical removal in patients with T1b/T2 esophagus cancer. Clinical trials are required.

## 1. Introduction

Esophageal cancer is the seventh most common cause of cancer-related death worldwide [1]. The current standard of care for treatment of T1b/T2 esophageal is esophagectomy [2,3]. Patients undergoing esophagectomy experience high rates of complications, disease recurrence, and postoperative mortality [4,5]. Esophagectomy is also followed by persistent deficits in physical function and increased symptom burden in areas such as dyspnea and pain [6]. Ultimately, this surgery exerts significant burdens on patients, caregivers, and healthcare systems with a high degree of healthcare resource utilization [7,8,9]. In contrast, a growing body of literature shows that endoscopic resection compared to esophagectomy for early-stage esophageal cancer is associated with fewer perioperative complications [10,11], decreased impact on quality of life [12], shorter length-of-stay [11,13], fewer readmissions [13], and superior all-cause and disease-specific mortality [11,14].

Endoscopic resection techniques have therefore become the standard of treatment for pT1a esophageal cancer [2]. In comparisons of esophagectomy to endoscopic resection alone for the treatment of pT1b esophageal cancer, esophagectomy demonstrates improved survival and lower recurrence rates; thus, it remains the standard of care for patients with pT1b and higher [10,11]. In patients who are ineligible for esophagectomy (due to medical comorbidity) or those who refuse esophagectomy after endoscopic resection identifies pT1b disease, there have been reports of adding adjuvant therapies like chemotherapy, radiation, or both in order to reduce the risk of locoregional and systemic recurrence [15,16]. However, a knowledge gap exists as to whether esophagectomy versus endoscopic resection with adjuvant therapy may result in improved patient outcomes among patients with T1b/T2 esophageal cancer. Filling this gap in the literature is important in order to avoid exposing patients to unnecessarily morbid procedures if organ-preserving alternatives exist. As such, the objective of this study was to determine whether endoscopic resection with adjuvant therapy versus esophagectomy results in improved outcomes among patients with pT1b/T2 esophageal cancer.

## 2. Materials and Methods

This is a systematic review and meta-analysis designed in accordance with the Preferred Reporting Items for Systematic Reviews and Meta-Analysis guidelines [17]. The full protocol was prospectively registered in the PROSPERO database (CRD42020193990). The search strategy was constructed using terms and subject headings related to esophageal cancer, endoscopic resection, and adjuvant therapy (Appendix A). The searched electronic databases are as follows: MEDLINE (OVID interface, 1946 onwards), EMBASE (OVID interface, 1974 onwards), Scopus (Elsevier interface, 1960 onwards), Cochrane Library (Wiley Interface, current issue), and CINAHL (EBSCOhost interface, 1954 onwards). These were queried at three timepoints: 5 July 2020, 2 June 2022, and 29 June 2024. Reference lists of selected studies and grey literature were also searched. There was no inception year designated.

The population of interest was adult patients with T1b or T2 esophageal cancer. The studied intervention was endoscopic mucosal resection (EMR) or endoscopic submucosal dissection (ESD) followed by any adjuvant therapy. The comparator of interest was esophagectomy. The single primary outcome was overall survival (OS). Secondary outcomes of interest included disease-free survival (DFS), cause-specific survival (CSS), occurrence of adverse events, local or metastatic recurrence, and perioperative mortality. If distinguishable from the data source, only major adverse events (i.e., Grade 3 or higher) were abstracted for the purposes of data analysis and synthesis.

Inclusion criteria were original research studies published in English of adult patients greater than 18 years old undergoing treatment for stage T1bN0 or T2N0 esophageal cancer with endoscopic resection plus adjuvant therapy. Studies were excluded for the following reasons: if they were not available with English translation, if they reported patients less than 18 years old, if patients underwent neoadjuvant therapy prior to endoscopic resection, if study patients were exclusively lower or higher stage than T1b/T2N0, or if the number of eligible patients undergoing endoscopic resection with adjuvant therapy was 5 or fewer. This threshold was arbitrarily defined so as to minimize the likelihood of including inappropriate case series while also permitting potential valid low-sample cohort studies to be included.

The literature search results were deduplicated and screened using Rayyan QCRI (Rayyan Systems Inc., Cambridge, MA, USA). The initial (July 2020) and first interval (June 2022) title/abstract screening and subsequent full-text review were conducted independently in duplicate by the first and second authors (EP and MR). The second interval (June 2024) title/abstract screening and full-text review were conducted independently in duplicate by the first and third authors (EP and NP). Disagreements were mediated by the senior author (BK) as required. Calibration was performed between reviewers during each phase of title/abstract and full-text screening after 10% of studies were randomly screened. Permissible Cohen’s kappa was defined as 0.61 or higher in order to proceed with the remainder of screening.

Data were extracted by MR into a pilot form for quality appraisal. Once a sample of three studies was entered into the pilot form, EP and MR reviewed the data. The form was amended to ensure efficient data extraction. After piloting, EP and MR extracted data independently up to 25% of included studies. At this point, duplicate check was conducted between 2 reviewers (EP and MR) to assess for discrepancies. Disagreements were mediated by BK as required. The remaining studies’ data, including demographic, clinical, and outcome data, were then extracted by a single reviewer (EP). Extracted data were recorded in Microsoft Excel version 16.94 and Cochrane Review Manager (RevMan) version 5.4. Risk-of-bias assessment was conducted independently by NP using the Cochrane Risk of Bias in Non-Randomized Studies of Interventions tool [18].

Descriptive statistics were collected to summarize study demographics, patient and tumour characteristics, and treatment outcomes. If studies combined patients undergoing endoscopic resection alone with endoscopic resection plus adjuvant therapy and did not report their outcomes separately, they were included in the quantitative analysis only if patients undergoing adjuvant therapy comprised greater than 50% of the sample. Studies that did not directly compare endoscopic resection with adjuvant therapy to esophagectomy were not included in the meta-analysis. Instead, these studies were included for descriptive purposes only. Hazard ratios from studies were abstracted for summary time-to-event analysis. If not available, hazard ratios were estimated in a hierarchical fashion using the methods described by Tierney et al., 2007 [19]. A fixed correction of −0.05% was applied to non-estimable time-to-event survival comparisons where one of two values was 100%, and −0.1% if both values were 100%. Using RevMan, summary time-to-event hazard ratios were calculated using the inverse variance method. Binary outcomes data were analyzed using the Mantel–Haenszel risk ratio. In accordance with the standard practice, continuity correction was deferred among non-estimable risk ratios with no events in either cohort because they would not provide an indication of either the direction or magnitude of the relative treatment effect [20]. Heterogeneity was assessed with the I² statistic. Random effects modelling was used. Subgroup analysis was performed on patients undergoing endoscopic resection according to majority resection type, majority adjuvant therapy type, and whether the majority of patients had lymphovascular invasion. Subgroup analysis was not performed if there were five or fewer studies in the pooled analysis or if all cohorts reported belonged to a single subgroup. Subgrouping by endoscopic resection was not performed on the recurrence outcome. This was performed to avoid the risk of confounding by indication because the standard of care for patients with higher-risk lesions is endoscopic submucosal dissection as opposed to endoscopic mucosal resection from the outset [21].

## 3. Results

A total of 1490 unique records were identified with the search strategy (Figure 1). A total of 84 full texts were screened, 33 of which were excluded. Overall, 51 studies were ultimately deemed eligible for inclusion [15,16,22,23,24,25,26,27,28,29,30,31,32,33,34,35,36,37,38,39,40,41,42,43,44,45,46,47,48,49,50,51,52,53,54,55,56,57,58,59,60,61,62,63,64,65,66,67,68,69,70]. Study designs included 48 retrospective cohort studies and 3 prospective cohort studies (Table 1). Publication dates ranged from 2003 to 2024 across seven different countries, the majority of which were published from patient populations in Japan (33/51). Study periods ranged from 1 year to 24 years. Risk-of-bias assessment identified a serious risk of bias in 48/51 studies (Table 2).

### 3.1. Summary of Included Studies

Demographic characteristics of patients were collected (Table 3). Sample sizes ranged from 10 to 759 patients with a median age ranging from 61 to 75 years. Tumour characteristics among patients in the included studies are shown in Table 4. The majority of tumour pathologic stages were pT1b. Most studies reported on tumours with predominantly squamous cell carcinoma histology (45/51). Types of endoscopic resection and adjuvant treatment among patients in the included studies are shown in Table 5. The majority of patients underwent chemoradiotherapy as the adjuvant therapy.

### 3.2. Primary Outcome

#### Five-Year Overall Survival

In 10 studies, 950 patients contributed to the 5-year OS outcome. As shown in Figure 2a, there was no significant difference in 5-year OS with combined endoscopic resection and adjuvant therapy versus esophagectomy (HR = 1.35; 95% CI = 0.74–2.44, *p* = 0.33). Subgroup analysis showed no significant difference in 5-year OS whether a majority of patients underwent ESD (HR = 1.33; 95% CI = 0.74–2.40, *p* = 0.34) or EMR (HR = 1.10; 95% CI = 0.16–7.55, *p* = 0.92). Subgroup analysis according to whether a majority of patients did or did not experience lymphovascular invasion showed no significant difference in 5-year OS between patients whose majority underwent chemoradiotherapy (HR = 0.90; 95% CI = 0.43–1.88, *p* = 0.78 and HR = 1.76; 95% CI = 0.75–4.15, *p* = 0.19, respectively). The single study that reported a majority of endoscopic resection patients undergoing ablative therapy (i.e., radiofrequency ablation) as adjuvant treatment demonstrated significantly worse 5-year OS compared to patients who underwent esophagectomy (HR = 3.73, 95% CI = 1.80–7.73, *p* = 0.0004); these patients were in a cohort where the majority did not have lymphovascular invasion. 

### 3.3. Secondary Outcomes

#### 3.3.1. Five-Year Disease-Free Survival

In six studies, 678 patients contributed to the 5-year DFS outcome. As shown in Figure 3, there was no significant difference in 5-year DFS with combined endoscopic resection and adjuvant therapy versus esophagectomy (HR = 1.12; 95% CI = 0.42–2.96, *p* = 0.82). Subgroup analysis according to whether a majority of patients did or did not have lymphovascular invasion showed no significant difference in 5-year DFS between patients who had adjuvant chemoradiotherapy (HR = 0.59; 95% CI = 0.21–1.65, *p* = 0.31 and HR = 1.94; 95% CI = 0.44–8.50, *p* = 0.38, respectively). The estimate of effect appears to favour endoscopic resection with adjuvant chemoradiotherapy in the subgroups with lymphovascular invasion, perhaps suggesting chemoradiotherapy benefits in lymphovascular invasion population who have a higher risk of developing locoregional recurrence. The single study that reported a majority of endoscopic resection patients undergoing ablative therapy as adjuvant demonstrated significantly worse 5-year DFS compared to patients who underwent esophagectomy (HR = 4.05, 95% CI = 1.66–9.87, *p* = 0.002); these patients were in a cohort where the majority did not have lymphovascular invasion. This was also the only study in the 5-year DFS analysis whose majority of patients underwent EMR as an endoscopic resection type. It is not possible to disentangle these two collinear variables.

#### 3.3.2. Five-Year Cause-Specific Survival

In five studies, 499 patients contributed to the 5-year cause-specific survival outcome. As shown in Figure 4, there was no significant difference in 5-year CSS with combined endoscopic resection and adjuvant therapy versus esophagectomy (HR = 2.98; 95% CI = 0.73–12.16, *p* = 0.13).

#### 3.3.3. Recurrence

In 15 studies that reported on recurrence and lymphovascular invasion, 1028 patients contributed to the recurrence outcome. As shown in Figure 5, there was no significant difference in the overall recurrence with combined endoscopic resection and adjuvant therapy versus esophagectomy (RR = 1.46; 95% CI = 0.75–2.83, *p* = 0.27). Subgroup analysis showed no significant difference based on whether the majority of patients did or did not have lymphovascular invasion (RR = 1.25; 95% CI = 0.47–3.31, *p* = 0.65 and RR = 1.75; 95% CI = 0.84–3.64, *p* = 0.13, respectively). In a sensitivity analysis, we added the patients from the two studies that did not report on lymphovascular invasion to the overall pooled analysis [34,39]; the results remained nonsignificant (RR = 1.38; 95% CI = 0.73–2.61, *p* = 0.32) (Appendix B).

#### 3.3.4. Adverse Events

In seven studies, 588 patients contributed to the adverse event outcomes. As shown in Figure 6, there was significantly fewer adverse events with combined endoscopic resection and adjuvant therapy as opposed to esophagectomy (RR = 0.65; 95% CI = 0.44–0.94, *p* = 0.02). Subgroup analysis showed no significant difference based on whether the majority of patients did or did not have lymphovascular invasion (RR = 0.75; 95% CI = 0.0.51–1.11, *p* = 0.15 and RR = 0.46; 95% CI = 0.19–1.10, *p* = 0.08, respectively). The majority of patients in all studies reporting adverse events underwent ESD.

#### 3.3.5. Perioperative Mortality

In 16 studies, 1090 patients contributed to the perioperative mortality outcome. Among the five estimable studies, threshold duration for perioperative mortality was defined only in Otaki et al., 2022 (30 days). As shown in Figure 7, there was no significant difference in perioperative mortality with combined endoscopic resection and adjuvant therapy versus esophagectomy (RR = 0.29; 95% CI = 0.07–1.14, *p* = 0.08), although the estimate of effect is highly skewed towards favouring endoscopic resection and adjuvant therapy.

### 3.4. Post Hoc Analysis

#### Endoscopic Resection with Adjuvant Therapy Versus Chemoradiotherapy Alone

Post hoc meta-analysis of the primary and secondary outcomes was performed on the studies comparing patients undergoing endoscopic resection with adjuvant therapy versus chemoradiotherapy alone (Appendix C). This was not included as an a priori aim during our trial protocol development; however, this became of interest during our analyses. There was no significant different in 5-year OS (HR = 0.29; 95% CI = 0.07–1.14, *p* = 0.08) (Figure A1), 5-year DFS (HR = 0.41; 95% CI = 0.17–1.00, *p* = 0.05) (Figure A2), 5-year CSS (HR = 0.58; 95% CI = 0.25–1.34, *p* = 0.20) (Figure A3), or adverse events (RR = 1.01; 95% CI = 0.81–1.27, *p* = 0.93) (Figure A4). Disease recurrence was significantly lower in the endoscopic resection with adjuvant therapy group (RR = 0.44; 95% CI = 0.28–0.69, *p* = 0.0004) (Figure A5).

Post hoc meta-analysis of the primary and secondary outcomes was performed based on subgroups according to tumour histology (Appendix D). Only one study in the meta-analysis that described a cohort with adenocarcinoma reported on the primary outcomes of interest. The single adenocarcinoma cohort significantly favoured esophagectomy for 5-year OS (HR = 3.73; 95% CI = 1.80–7.73, *p* = 0.0004) and 5-year DFS (HR = 4.05, 95% CI = 1.66–9.87, *p* = 0.002). The 5-year OS and DFS did not favour either esophagectomy or endoscopic resection with adjuvant therapy in the squamous cell carcinoma subgroup. There were no differences in recurrence or perioperative mortality according to adenocarcinoma versus squamous cell carcinoma subgroups. Because no studies that reported on 5-year CSS or adverse events included patients with adenocarcinoma, subgroup analysis based on histology could not be performed on these outcomes.

## 4. Discussion

Using a systematic review and meta-analysis of the literature, studies comparing endoscopic resection with adjuvant therapy versus esophagectomy among patients with pT1b/T2 esophageal cancer demonstrated no significant difference in 5-year overall survival. Secondary measures such as 5-year DFS, 5-year CSS, recurrence, and perioperative mortality also showed no significant differences. Patients undergoing endoscopic resection with adjuvant therapy appear to experience fewer major adverse events. Esophagectomy showed favourable 5-year OS and DFS only among patients in a single study where the majority of patients underwent ablation as adjuvant therapy, suggesting that purely ablative adjuvant therapy may be an insufficient adjuvant therapy even in a low-risk disease without the evidence of lymphovascular invasion. Importantly, the results are limited in the conclusions that can be drawn due to their high risk of bias and heterogeneity of patient characteristics in areas such as tumour characteristics and treatment received.

### 4.1. Comparisons to Previous Reports

Compared to a previous systematic review and meta-analysis examining endoscopic resection with adjuvant therapy versus esophagectomy for T1a/T1b esophageal cancers, Chen et al., 2023, also found no significant difference in OS between groups [71]. In contrast, these authors determined improved DFS in the esophagectomy group, whereas our study shows no difference despite our inclusion of only higher-risk disease (i.e., pT1b and above). Many of the differences in our review herein and the review published by Chen et al., 2023, are methodological. First, it is unclear at which timepoint (i.e., 1-year, 3-year, etc.) these time-to-event measures were assessed in the previous systematic review. Second, our review includes patients with pT1b/T2 disease, whereas the review by Chen et al., 2023, includes patients with T1a/T1b disease. Third, Chen et al., 2023, excluded studies based on the comparison group and outcome measures. However, our review did not exclude studies on this basis. Together, these three factors may account for the difference in reported DFS, combined with our inclusion of more recent studies after 2023. Overall, the patients included in our analysis were at a higher risk. Furthermore, the different inclusion criteria likely resulted in differences in the quantity and types of studies included in our respective pooled analyses.

When compared to the two most recent systematic review and meta-analyses of endoscopic resection alone versus esophagectomy for early esophageal cancer, these studies also found significantly fewer major adverse events among patients undergoing endoscopic resection [10,11]. The lower perioperative mortality rate for endoscopic resection identified by Zheng et al., 2021, was not observed in our meta-analysis. This is likely spurious due to the fact that the majority of studies reported no perioperative mortality in either group, thus precluding estimable risk ratios. Nevertheless, the visual inspection of our individual point estimates as well as our forest plot in Figure 7 and the overall estimate of effect suggest that there is a signal of improved perioperative mortality outcomes with endoscopic resection (RR = 0.29; 95% CI = 0.07–1.14, *p* = 0.08).

The significantly reduced 5-year OS and DFS among the majority of the ablation cohort reported in Otaki et al., 2020, are not unexpected [36]. Evidence comparing ablation as adjuvant therapy versus esophagectomy is limited to intramucosal lesions only, not pT1b/T2 lesions [72]. Currently, there is only conditional recommendation for its use after endoscopic resection for Barret’s esophagus with high-grade dysplasia [73]. These, combined with our results, support CRT as the adjuvant therapy for pT1b/T2 lesions instead of ablation given these lesions will have a propensity towards a higher risk for locoregional recurrence. More importantly, our results further demonstrate that, regardless of whether patients experienced lymphovascular invasion, endoscopic resection with adjuvant therapy demonstrated equal 5-year OF and DFS compared to esophagectomy. This is despite the lymphovascular invasion that places patients at a significantly higher risk of metastatic disease. As such, even among patients with high-risk pT1b disease, organ-sparing therapy could be a valid option instead of radical resection that would afford similar long-term survival outcomes.

### 4.2. Clinical Relevance

When placed among the broader literature, the current standard of care for pT1b/T2N0 esophageal cancer is esophagectomy [3]. In those suspected to have T2N0 disease, patients may also receive neoadjuvant chemoradiotherapy according to the CROSS protocol [74,75]. Patients with clinical T1 disease that undergo endoscopic resection and then are discovered to have pT1b or pT2 disease are currently recommended to undergo esophagectomy. For patients that are ineligible for or who decline esophagectomy, the current first-line treatment is chemoradiation or subsequent surveillance for recurrence, metastatic disease, and possible transition to palliative therapy. If unable to tolerate chemoradiation, patients may be directed to palliative radiotherapy from the outset [3]. Our study shows that there may be an alternative curative option available to patients that are ineligible for or do not wish to have esophagectomy. This includes patients deemed ineligible because of inoperability due to comorbidities that place them at an unacceptable perioperative risk or due to patient preference. The previous literature shows endoscopic resection has a lower impact on quality of life compared to esophagectomy among esophageal cancer survivors [12]. As such, we also anticipate that endoscopic resection with adjuvant therapy would be preferable for patients wishing to preserve quality of life. Furthermore, our post hoc analysis of studies with cohorts comparing endoscopic resection with adjuvant therapy to definitive chemoradiation alone demonstrates equivalent survival outcomes and incidence of adverse events but with the apparent added benefit of lower disease recurrence among patients receiving endoscopic resection followed by adjuvant therapy. Therefore, endoscopic resection with adjuvant therapy may also be preferable to definitive chemoradiation to some patients. Post hoc subgroup analysis suggests that esophagectomy may be favoured in the sole study that explored this question in adenocarcinoma patients; however, the adjuvant therapy in Otaki et al., 2020, was predominantly ablation, which in multiple subgroups has been identified as conferring lower survival and theoretically also can be understood to confer poor locoregional control as opposed to adjuvant therapy or chemotherapy. Thus, this adenocarcinoma subgroup is highly unreliable.

### 4.3. Limitations

Due to the lack of available RCT-level evidence, this analysis contains a high risk of bias because it is derived exclusively from cohort studies. This imparts a serious degree of confounding and selection bias to all of our included studies. Major variables that may determine the treatment pathway of this patient population such as medical comorbidity and quality of life were not well reported as others. As such, it is likely that there is a selection phenomenon at play that was not adequately captured. Therefore, clinical interpretation should be limited. Our search included publications only in English, which may have inadvertently excluded eligible studies. Because the majority of included studies described patients with squamous cell carcinoma, our results are limited in their applicability to patients who experience adenocarcinoma. Therefore, the generalizability of our results is limited. Furthermore, the reporting of adverse events among included studies was heterogeneous. Due to this lack of standardized adverse events reporting, it is possible that major adverse events were not comparable between endoscopic resection and esophagectomy groups. The manner in which the intervention groups were separated in outcome reporting was also heterogenous. While the outcomes for the majority of patients undergoing ER with adjuvant therapy were distinguished from ER alone among the majority of included studies, this was not the case in two of the studies included in the meta-analysis. Ota et al., 2003, and Otaki et al., 2020, described a single group of patients with and without adjuvant therapy where 72.2% and 67.1% received adjuvant therapy, respectively. As such, there is likely more favourable pathology among these patient populations that could subvert the interpretation of our results.

In terms of outcomes, it is likely that patients in real-world scenarios undergoing esophagectomy only receive clinical staging and risked being understaged. As such, the applicability of these survival outcomes to patients who are candidates for esophagectomy may be subverted. Moreover, there was a lack of patient-reported outcomes and health-related quality of life measures. However, this is a limitation of the existing data rather than a limitation of this review per se. Lastly, our analysis was not risk-adjusted. Despite this limitation, we hypothesize that risk adjustment would show better results for endoscopic resection and adjuvant therapy. This is because patients who are receiving endoscopic resection and adjuvant therapy rather than the standard-of-care esophagectomy are precisely those who had such prohibitive medical comorbidities that they were deemed to be too high risk for esophagectomy. These are the patients that would be expected to have a higher risk of complications with any intervention and also a lower chance of achieving long-term survival, even ignoring their esophageal cancer. Thus, adjusting for a prognostic risk in a risk-adjusted prospective comparative study may demonstrate even better outcomes than those currently reported in our study.

Overall, the studies included are highly heterogeneous in a variety of domains. Therefore, this limits the immediate clinical applicability of our study. This heterogeneity is due to the nature of the evidence that is currently available to us. Therefore, we have attempted to be inclusive of this reality. We suspect that residual bias due to heterogeneity is operating in favour of esophagectomy as opposed to endoscopic resection at this time. While we are unable to make clinical recommendations based on these results, they do suggest equipoise to substantiate clinical trials that could control for the heterogeneity identified herein.

### 4.4. Strengths

The strength of this study is that we have identified similarity in 5-year overall survival between patients with pT1b/T2 esophageal cancer undergoing endoscopic resection plus adjuvant therapy compared to esophagectomy. To our knowledge, this also represents the first systematic review and meta-analysis comparing endoscopic resection with adjuvant therapy to esophagectomy that includes patients with higher risk pT1b and pT2 disease. We adhered to a prospectively registered protocol according to the established criteria and defined an a priori analysis plan. The search strategy and inclusion criteria were constructed to maximize sensitivity of the search, thereby reducing the likelihood that eligible studies were overlooked. All screening was conducted independently in duplicate with pre-planned calibration exercises at regular intervals to improve reviewer concordance.

## 5. Conclusions

Although esophagectomy remains the standard of care for patients discovered to have pT1b/T2 esophageal cancer, this systematic review suggests that endoscopic resection followed by adjuvant therapy may be a possible alternative to esophagectomy. This is supported by multiple equivalent 5-year survival measures, similar disease recurrence, and fewer adverse events among patients receiving endoscopic resection followed by adjuvant therapy. However, our conclusions are limited to patients with squamous cell carcinoma. The applicability to patients with adenocarcinoma remains unclear. Unfortunately, patient-reported outcomes are also neglected amongst the available literature despite playing a crucial role in the decision-making process as well. Overall, this heterogeneity in both patient characteristics and outcomes reporting significantly limits the clinical applicability of our findings. Therefore, higher-level evidence is needed to substantiate the findings reported. The findings of this study suggest that there may be enough equipoise to support a randomized controlled trial to fully answer this question. Patient-reported outcomes such as quality of life should be included. Currently, a feasibility trial of this kind is underway [76]. For patients who seek organ preservation and minimal risk of adverse events, endoscopic resection with adjuvant therapy may be preferable.

## Figures and Tables

**Figure 1 cancers-17-00680-f001:**
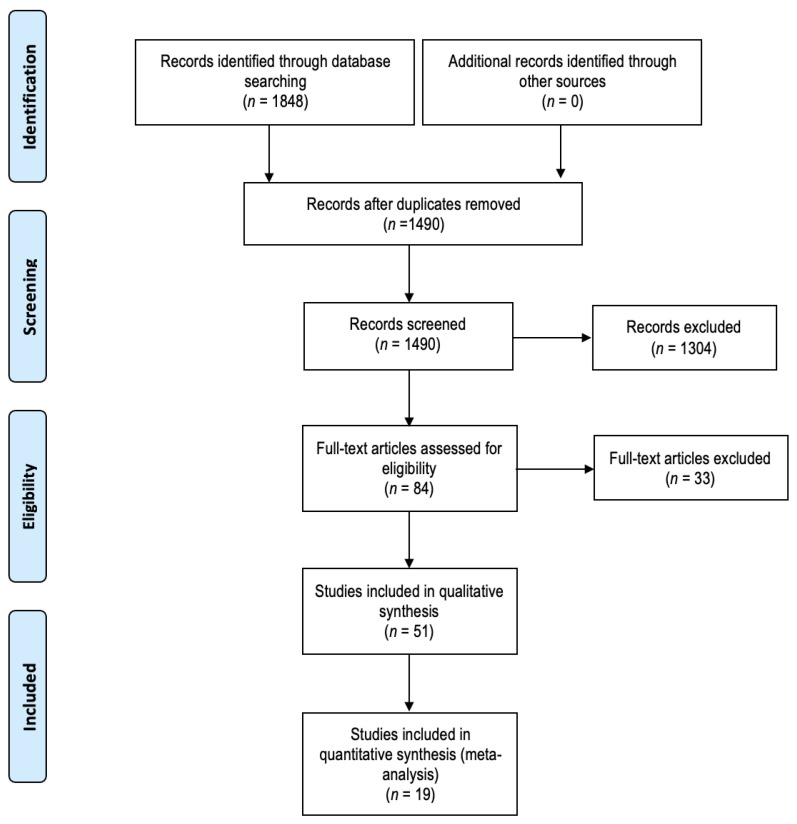
Number of records identified, included, and excluded through the different phases of the systematic review, adapted from the Preferred Reporting Items for Systematic Reviews and Meta-Analyses 2009 flow diagram.

**Figure 2 cancers-17-00680-f002:**
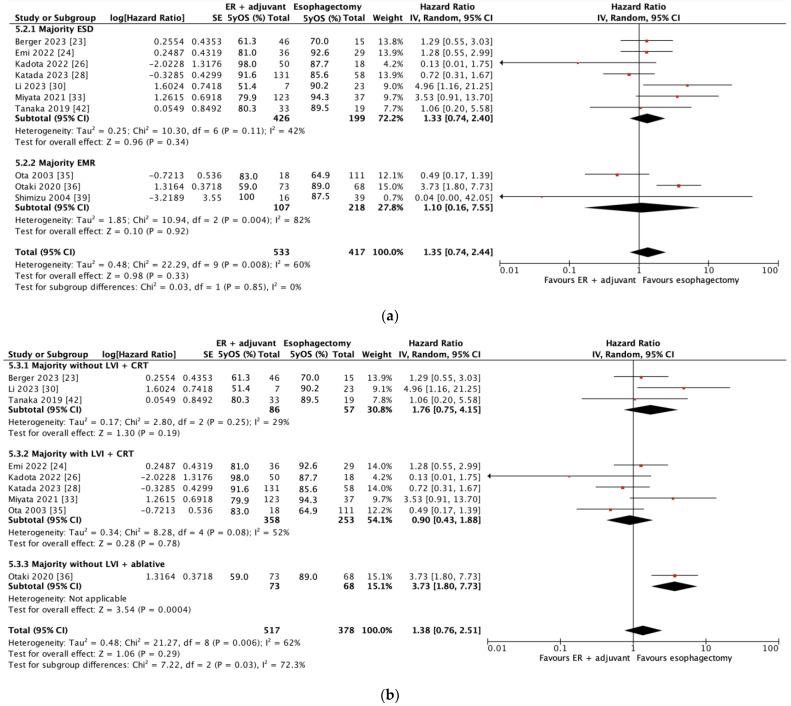
(**a**) Forest plot of combined 5-year overall survival and according to subgroups by endoscopic resection type (data are depicted by using hazard ratios with 95% confidence intervals). CI = confidence interval; ER = endoscopic resection; EMR = endoscopic mucosal resection; ESD = endoscopic submucosal dissection; IV = inverse variance; 5yOS = 5-year overall survival. (**b**) Forest plot of 5-year overall survival based on subgroups by whether majority of patients experienced lymphovascular invasion and majority adjuvant therapy type (data are depicted by using hazard ratios with 95% confidence intervals). CI = confidence interval; CRT = chemoradiotherapy; ER = endoscopic resection; IV = inverse variance; LVI = lymphovascular invasion; 5yOS = 5-year overall survival. [23,24,26,28,30,33,35,36,39,42].

**Figure 3 cancers-17-00680-f003:**
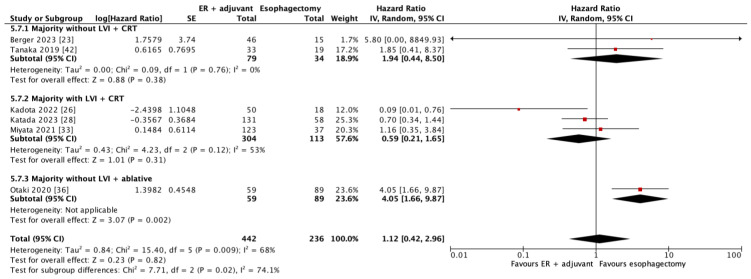
Forest plot of 5-year disease-free survival (data are depicted by using hazard ratios with 95% confidence intervals). CI = confidence interval; CRT = chemoradiotherapy; ER = endoscopic resection; IV = inverse variance; LVI = lymphovascular invasion. [23,26,28,33,36,42].

**Figure 4 cancers-17-00680-f004:**
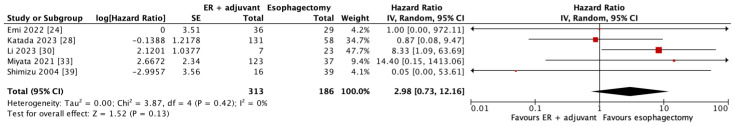
Forest plot of 5-year cause-specific survival (data are depicted by using hazard ratios with 95% confidence intervals). CI = confidence interval; ER = endoscopic resection; IV = inverse variance. [24,28,30,33,39].

**Figure 5 cancers-17-00680-f005:**
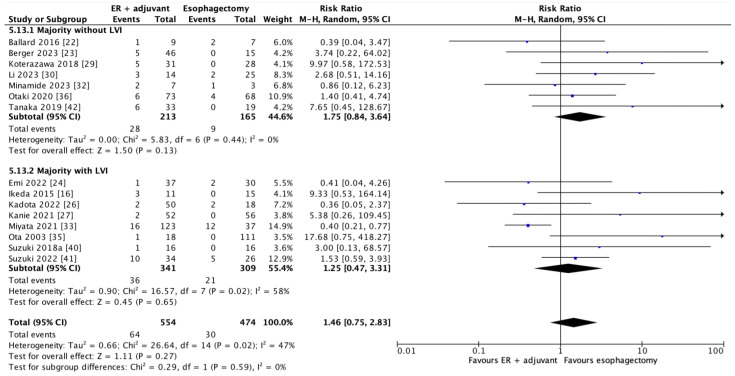
Forest plot of overall recurrence and based on subgrouping by lymphovascular invasion (data are depicted by using hazard ratios with 95% confidence intervals). CI = confidence interval; IV = inverse variance; M-H = Mantel–Haenszel. [16,22,23,24,26,27,29,30,32,33,35,36,40,41,42].

**Figure 6 cancers-17-00680-f006:**
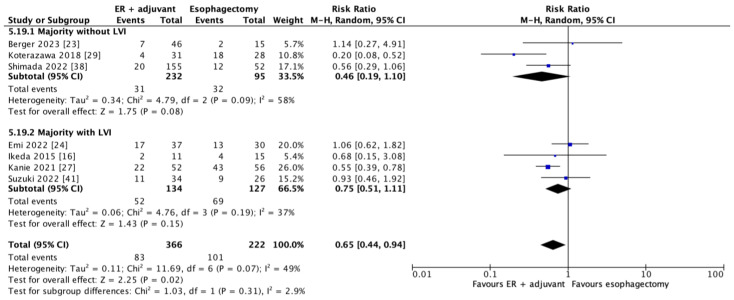
Forest plot of overall adverse events and based on subgrouping by lymphovascular invasion (data are depicted by using hazard ratios with 95% confidence intervals). CI = confidence interval; IV = inverse variance; M-H = Mantel–Haenszel. [16,23,24,27,29,38,41].

**Figure 7 cancers-17-00680-f007:**
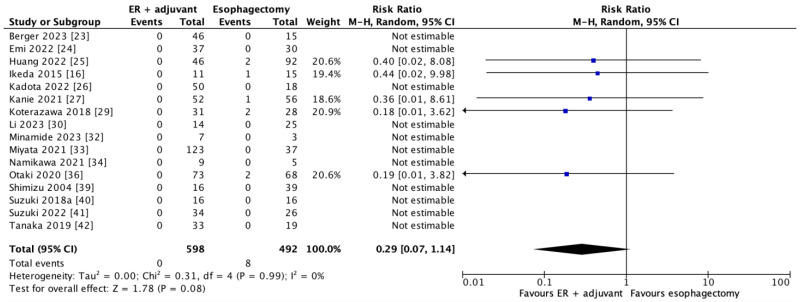
Forest plot of secondary outcome of perioperative mortality (data are depicted by using risk ratios with 95% confidence intervals). CI = confidence interval; M-H = Mantel–Haenszel. [16,23,24,25,26,27,29,30,32,33,34,36,39,40,41,42].

**Table 1 cancers-17-00680-t001:** Characteristics of studies eligible for inclusion.

Study	Country	Design	Period	Number of Centres (*n*)	Median Months Follow-Up (ER + Adjuvant, Esophagectomy)
ER + adjuvant (>50%) vs. esophagectomy					
Ballard 2016 [22]	United States	Retrospective cohort	2001–2013	Multiple (2)	27, 49
Berger 2023 [23]	France	Retrospective cohort	April 1999–April 2018	Multiple (11)	22, 22
Emi 2022 [24]	Japan	Retrospective cohort	January 2000–June 2017	Single	60, 60 *
Huang 2022 [25]	United States, China	Retrospective cohort	September 2011–February 2017	Multiple (9)	32, 32
Ikeda 2015 [16]	Japan	Retrospective cohort	January 2005–December 2010	Single	43, 47
Kadota 2022 [26]	Japan	Retrospective cohort	January 2009–December 2017	Single	60.6, 60.6
Kanie 2021 [27]	Japan	Retrospective cohort	2005–2019	Single	54, 54
Katada 2023 [28]	Japan	Retrospective cohort	August 1992–April 2016	Multiple (9)	58.6, 58.6
Koterazawa 2018 [29]	Japan	Retrospective cohort	2005–2016	Single	41, 45
Li 2023 [30]	China	Retrospective cohort	January 2012–July 2019	Multiple (2)	57.8, 57.8
Minamide 2023 [32]	Japan	Retrospective cohort	June 2009–June 2020	Single	47.7, 47.7
Miyata 2021 [33]	Japan	Retrospective cohort	January 2006–August 2018	Single	60.3, 60.3
Namikawa 2021 [34]	Japan	Retrospective cohort	July 2010–January 2015	Single	58.6, 58.6
Ota 2003 [35]	Japan	Retrospective cohort	1992–2000	Single	27.8, -
Otaki 2020 [36]	United States	Retrospective cohort	October 2001–October 2016	Multiple (3)	43.4, 49.4
Rodriguez de Santiago 2024 [37]	Spain	Retrospective cohort	January 2016–December 2021	Multiple (registry/database)	14, 14
Shimada 2022 [38]	Japan	Retrospective cohort	January 2006–August 2017	Multiple (21)	67, 67
Shimizu 2004 [39]	Japan	Prospective cohort	October 1996–January 2002	Multiple (2)	43, 38
Suzuki 2018a [40]	Japan	Retrospective cohort	January 2014–April 2017	Single	24, 24
Suzuki 2022 [41]	Japan	Retrospective cohort	January 2008–December 2021	Single	46, 56
Tanaka 2019 [42]	Japan	Retrospective cohort	January 2002–December 2013	Single	60, 60
Yuan 2024 [43]	China	Retrospective cohort	2017	Single	60.2, 60.9
ER + adjuvant (<50%) vs. esophagectomy					
Arima 2007 [44]	Japan	Prospective cohort	March 2000–September 2006	Single	32, 32
Matsueda 2022 [31]	Japan	Retrospective cohort	January 2008–December 2016	Single	67.4, 67.4
Noordzij 2023 [45]	Netherlands	Retrospective cohort	January 2000–December 2014	Multiple (registry/database)	-, -
Qian 2022 [46]	China	Retrospective cohort	January 2015–December 2021	Single	30, 28
Wang 2022 [47]	Taiwan	Retrospective cohort	January 2008–December 2016	Single	49.2, 50.9
Zhang 2024 [48]	China	Retrospective cohort	March 2009–May 2021	Single	47.3, 57.8
ER + adjuvant vs. CT/CRT/RT alone					
Kaburagi 2012 [49]	Japan	Retrospective cohort	January 2000–December 2009	Single	76.6, 76.6
Kawaguchi 2015 [50]	Japan	Retrospective cohort	October 2000–December 2011	Single	39, 34.2
Lyu 2022 [51]	China	Retrospective cohort	January 2011–June 2021	Single	51.9, 51.9
Minashi 2019 [52]	Japan	Prospective cohort	December 2006–July 2012	Multiple (23)	60, 60
Miyazaki 2022 [53]	Japan	Retrospective cohort	2004–2007	Single	59.8, 59.8
Suzuki 2018b [54]	Japan	Retrospective cohort	June 2009–September 2017	Single	33, 33
Uchinami 2016 [55]	Japan	Retrospective cohort	2004–2011	Single	43.6, 43.6
Yoshimizu 2018 [56]	Japan	Retrospective cohort	April 2003–December 2014	Single	72, 62
ER + adjuvant vs. ER alone					
Chen 2019 [57]	United States	Retrospective cohort	1998–2013	Multiple (registry/database)	-, -
Dermine 2020 [58]	France	Retrospective cohort	2015–2020	Multiple (2)	28, 27
Hisano 2018 [59]	Japan	Retrospective cohort	March 2005–December 2014	Single	36, 36
Konishi 2024 [15]	Japan	Retrospective cohort	September 2007–December 2019	Single	71.1, 71.1
Ren 2022 [60]	China	Retrospective cohort	January 2015–December 2019	Single	36
Suzuki 2021 [61]	Japan	Retrospective cohort	January 2010–April 2019	Single	74.1, 57.4 **
Yang 2023 [62]	China	Retrospective cohort	January 2010–December 2019	Multiple (11)	53.4, 53.4
ER + adjuvant alone					
Cho 2024 [63]	Japan	Retrospective cohort	August 2007–December 2017	Single	71
Greenwald 2010 [64]	United States	Retrospective cohort	2006–2009	Multiple (10)	10.6 **
Hamada 2017 [65]	Japan	Retrospective cohort	January 2006–December 2012	Single	51
Ikawa 2019 [66]	Japan	Retrospective cohort	January 2006–December 2014	Single	61
Mochizuki 2011 [67]	Japan	Retrospective cohort	November 2004–June 2010	Single	45
Nemoto 2005 [68]	Japan	Retrospective cohort	May 1996–October 2002	Multiple (4)	33
Nishibuchi 2022 [69]	Japan	Retrospective cohort	2011–2018	Single	48
Yang 2021 [70]	China	Retrospective cohort	January 2010–August 2019	Single	35.5

ER + adjuvant vs. esophagectomy studies were divided into two groups denoting whether patients undergoing adjuvant therapy comprised greater or fewer than 50% of the patients undergoing ER. ER = endoscopic resection; CT = chemotherapy; CRT = chemoradiotherapy; RT = radiotherapy; “-” = not reported. * Minimum 5-year follow-up. ** Mean reported instead of median.

**Table 2 cancers-17-00680-t002:** Risk-of-bias assessment for studies eligible for inclusion.

Study	Confounding	Selection of Participants	Classification of Interventions	Deviation from Intended Interventions	Missing Data	Measurement of Outcomes	Selection of Reported Result	Overall Risk of Bias Judgment
ER + adjuvant (>50%) vs. esophagectomy								
Ballard 2016 [22]	Serious	Serious	Low	Low	Low	Low	Low	Moderate
Berger 2023 [23]	Serious	Serious	Low	Low	Low	Low	Low	Serious
Emi 2022 [24]	Serious	Serious	Low	Low	Low	Low	Low	Serious
Huang 2022 [25]	Serious	Serious	Low	Low	Low	Low	Low	Serious
Ikeda 2015 [16]	Serious	Serious	Low	Low	Low	Low	Low	Serious
Kadota 2022 [26]	Serious	Serious	Low	Low	Moderate	Low	Low	Serious
Kanie 2021 [27]	Serious	Serious	Low	Low	Low	Low	Low	Serious
Katada 2023 [28]	Serious	Serious	Low	Low	Low	Low	Low	Serious
Koterazawa 2018 [29]	Serious	Serious	Low	Low	Low	Low	Low	Serious
Li 2023 [30]	Serious	Serious	Low	Low	Low	Low	Low	Serious
Minamide 2023 [32]	Serious	Serious	Low	Low	Low	Low	Low	Serious
Miyata 2021 [33]	Serious	Serious	Low	Low	Low	Moderate	Low	Serious
Namikawa 2021 [34]	Serious	Serious	Low	Low	Low	Low	Low	Moderate
Ota 2003 [35]	Serious	Serious	Low	Low	Low	Low	Low	Serious
Otaki 2020 [36]	Serious	Serious	Low	Low	Low	Low	Low	Serious
Rodriguez de Santiago 2024 [37]	Serious	Serious	Low	Low	Low	Low	Low	Serious
Shimada 2022 [38]	Serious	Serious	Low	Low	Low	Low	Low	Serious
Shimizu 2004 [39]	Serious	Serious	Low	Low	Low	Low	Low	Serious
Suzuki 2018a [40]	Serious	Serious	Low	Low	Low	Low	Low	Serious
Suzuki 2022 [41]	Serious	Serious	Low	Low	Low	Low	Low	Serious
Tanaka 2019 [42]	Serious	Serious	Low	Low	Low	Low	Low	Serious
Yuan 2024 [43]	Serious	Serious	Low	Low	Low	Low	Low	Serious
ER + adjuvant (<50%) vs. esophagectomy								
Arima 2007 [44]	Serious	Serious	Low	Low	Moderate	Low	Low	Serious
Matsueda 2022 [31]	Serious	Serious	Low	Low	Low	Low	Low	Serious
Noordzij 2023 [45]	Serious	Serious	Low	Low	Low	Low	Low	Serious
Qian 2022 [46]	Serious	Serious	Low	Low	Low	Low	Low	Serious
Wang 2022 [47]	Serious	Serious	Low	Low	Low	Low	Low	Serious
Zhang 2024 [48]	Serious	Serious	Low	Low	Low	Low	Low	Serious
ER + adjuvant vs. CT/CRT/RT alone								
Kaburagi 2012 [49]	Serious	Serious	Low	Low	Low	Low	Low	Serious
Kawaguchi 2015 [50]	Serious	Serious	Low	Low	Low	Low	Low	Serious
Lyu 2022 [51]	Serious	Serious	Low	Low	Low	Low	Low	Serious
Minashi 2019 [52]	Serious	Serious	Low	Low	Low	Low	Low	Serious
Miyazaki 2022 [53]	Serious	Serious	Low	Low	Low	Low	Low	Moderate
Suzuki 2018b [54]	Serious	Serious	Low	Low	Low	Low	Low	Serious
Uchinami 2016 [55]	Serious	Serious	Low	Low	Low	Low	Low	Serious
Yoshimizu 2018 [56]	Serious	Serious	Low	Low	Low	Low	Low	Serious
ER + adjuvant vs. ER alone								
Chen 2019 [57]	Serious	Serious	Low	Low	Low	Low	Low	Serious
Dermine 2020 [58]	Serious	Serious	Low	Low	Low	Low	Low	Serious
Hisano 2018 [59]	Serious	Serious	Low	Low	Low	Moderate	Low	Serious
Konishi 2024 [15]	Serious	Serious	Low	Low	Low	Low	Low	Serious
Ren 2022 [60]	Serious	Serious	Low	Low	Low	Low	Low	Serious
Suzuki 2021 [61]	Serious	Serious	Low	Low	Low	Low	Low	Serious
Yang 2023 [62]	Serious	Serious	Low	Low	Low	Low	Low	Serious
ER + adjuvant alone								
Cho 2024 [63]	Serious	Serious	Low	Low	Low	Low	Low	Serious
Greenwald 2010 [64]	Serious	Serious	Low	Low	Low	Low	Low	Serious
Hamada 2017 [65]	Serious	Serious	Low	Low	Low	Low	Low	Serious
Ikawa 2019 [66]	Serious	Serious	Low	Low	Low	Low	Low	Serious
Mochizuki 2011 [67]	Serious	Serious	Low	Low	Low	Low	Low	Serious
Nemoto 2005 [68]	Serious	Serious	Low	Low	Low	Low	Low	Serious
Nichibuchi 2022 [69]	Serious	Serious	Low	Low	Low	Low	Low	Serious
Yang 2021 [70]	Serious	Serious	Low	Low	Low	Low	Low	Serious

ER + adjuvant vs. esophagectomy studies were divided into two groups denoting whether patients undergoing adjuvant therapy comprised greater or fewer than 50% of the patients undergoing ER. ER = endoscopic resection; CT = chemotherapy; CRT = chemoradiotherapy; RT = radiotherapy.

**Table 3 cancers-17-00680-t003:** Patient demographic characteristics among included studies.

Study	*n*		Female Gender, *n* (%)		Median Age (Range)		Median CCI (Range)	
	ER + Adjuvant	Comparator	ER + Adjuvant	Comparator	ER + Adjuvant	Comparator	ER + Adjuvant	Comparator
ER + adjuvant (>50%) vs. esophagectomy								
Ballard 2016 [22]	9	7	-	-	70 ± 14 *	62 ± 5 *	-	-
Berger 2023 [23]	46	15	10 (21.7)	4 (28.6)	63.8 (35–83) *	60.0 (44–81) *	-	-
Emi 2022 [24]	37	30	3 (8.1)	5 (16.7)	66 (57–81)	62 (54–78)	-	-
Huang 2022 [25]	46	92	12 (26.1)	21 (22.8)	61.1 ± 8.1 *	61.1 ± 6.7 *	-	-
Ikeda 2015 [16]	11	15	6 (14.0) **		69 (49–79)	68 (47–72)	4 (-)	2 (-)
Kadota 2022 [26]	50	18	11 (22.0)	2 (11.1)	68 (57–79)	68.5 (46–82)	-	-
Kanie 2021 [27]	52	56	6 (11.5)	11 (19.6)	65.0 (50–77)	61.5 (46–79)	0 (0–6)	1 (0–4)
Katada 2023 [28]	131	58	-	-	-	-	-	-
Koterazawa 2018 [29]	31	28	7 (22.6)	4 (14.3)	68 (50–81)	66 (47–77)	-	-
Li 2023 [30]	14	25	16 (41.0) **		61.4 ± 7.4 *		-	-
Minamide 2023 [32]	7	3	1 (10.0) **		73.0 (-) **		80% 0–1, 20% ≥2 **	
Miyata 2021 [33]	123	37	10 (8.1)	6 (16.2)	66.9 ± 8.9 *	59.9 ± 11.7*	-	-
Namikawa 2021 [34]	9	5	10 (15.1) **		67.2 (39–81) **		-	-
Ota 2003 [35]	18 **	111	1 (5.6)	-	67.9 (47–86)	-	-	-
Otaki 2020 [36]	73	68 ***	17 (23.3)	5 (7.4)	73.4 (-)	64.1 (-)	5 (-)	4 (-)
Rodriguez de Santiago 2024 [37]	11	9	-	-	-	-	-	-
Shimada 2022 [38]	155	52	-	-	-	-	-	-
Shimizu 2004 [39]	16	39	1 (6.3)	3 (7.7)	62.5 ± 7.8 *	63.3 ± 8.4 *	-	-
Suzuki 2018a [40]	16	16	3 (18.8)	0 (0.0)	67 (46–87)	64 (51–77)	-	-
Suzuki 2022 [41]	34	26	5 (14.7)	2 (7.7)	69 (50–80)	65 (45–78)	-	-
Tanaka 2019 [42]	33	19	1 (3.0)	4 (21.1)	65 (50–79)	63 (43–76)	0 (0–4)	0 (0–5)
Yuan 2024 [43]	8	273	-	-	-	-	-	-
ER + adjuvant (<50%) vs. esophagectomy								
Arima 2007 [44]	15	60	1 (7.1)	7 (11.7)	64 (53–75)	64 (44–85)	-	-
Matsueda 2022 [31]	91	15	-	-	-	-	-	-
Noordzij 2023 [45]	14	351	-	-	-	-	-	-
Qian 2022 [46]	10	202	-	62 (30.7)		64.8 ± 8.19 *	-	-
Wang 2022 [47]	7	32	-	0 (0.0)	-	54 (40–72)	-	-
Zhang 2024 [48]	6	73 ***	-	19 (26.0)	-	64.38 ± 7.64 *	-	0 (0–1)
ER + adjuvant vs. CT/CRT/RT alone								
Kaburagi 2012 [49]	7	9	-	-	-	-	-	-
Kawaguchi 2015 [50]	16	31	1 (6.3)	6 (19.4)	65 (42–77)	68 (33–80)	-	-
Lyu 2022 [51]	18	37	4 (22.2)	12 (32.4)	62.0 (51–74)	69.0 (52–83)	-	-
Minashi 2019 [52]	83	13	-	-	-	-	-	-
Miyazaki 2022 [53]	63	28	-	-	-	-	-	-
Suzuki 2018b [54]	29	21	5 (17.2)	4 (19.0)	68 (50–82)	75 (59–87)	-	-
Uchinami 2016 [55]	45	26	14 (19.7) **		70 (47–84) **		-	-
Yoshimizu 2018 [56]	21	43	1 (4.8)	1 (2.3)	66 (50–79)	71 (52–83)	-	-
ER + adjuvant vs. ER alone								
Chen 2019 [57]	88	671	21 (23.9)	141 (21.0)	70.4 ± 6.2 *	69.5 ± 7.0 *	-	-
Dermine 2020 [58]	28	13	3 (10.7)	2 (15.4)	65 (53–85)	61 (47–78)	-	-
Hisano 2018 [59]	13	14	3 (23.1)	2 (14.3)	66 ± 5 *	68 ± 7.4 *	-	-
Konishi 2024 [15]	48	333	-	-	-	-	-	-
Ren 2022 [60]	41	13	10 (35.7)	6 (46.2)	67 (38–87)	74 (61–91)	-	-
Suzuki 2021 [61]	64	82	5 (7.8)	8 (9.8)	66.6 ± 9.0 *	70.6 ± 9.4	-	-
Yang 2023 [62]	47	114	11 (23.4)	23 (20.2)	-	-	-	-
ER + adjuvant alone								
Cho 2024 [63]	73	N/A	12 (16.4)	N/A	67 (52–78)	N/A	-	N/A
Greenwald 2010 [64]	27	N/A	8 (11.4) **	N/A	75 (51–88) **	N/A	-	N/A
Hamada 2017 [65]	66	N/A		N/A	67 (45–82)	N/A	-	N/A
Ikawa 2019 [66]	96	N/A	5 (7.3)	N/A	67 (42–82)	N/A	-	N/A
Mochizuki 2011 [67]	14	N/A	0 (0.0)	N/A	65 (49–78) *	N/A	-	N/A
Nemoto 2005 [68]	30	N/A	4 (13.3)	N/A	68 (53–82)	N/A	-	N/A
Nichibuchi 2022 [69]	50	N/A	3 (6.0)	N/A	68 (54–81)	N/A	-	N/A
Yang 2021 [70]	31	N/A	6 (19.4)	N/A	62 (49–78)	N/A	-	N/A

ER + adjuvant vs. esophagectomy studies were divided into two groups denoting whether patients undergoing adjuvant therapy comprised greater or fewer than 50% of the patients undergoing ER. CCI = Charlson Comorbidity Index; ER = endoscopic resection. CT = chemotherapy; CRT = chemoradiotherapy; N/A = not applicable; RT = radiotherapy; “-” = not reported. * Mean and/or standard deviation reported instead of median and/or range. ** Not reported separately from other groups. *** 5/68 (Otaki 2020) and 16/72 (Zhang 2024) also received adjuvant chemoradiotherapy.

**Table 4 cancers-17-00680-t004:** Tumour characteristics among included studies.

Study	pTNM (*n*)		Majority Histology (%)		Median Tumour Length, Millimetres (Range)		Tumour Location (Ce/Ut/Mt/Lt/Ae)		Lymphovascular Invasion, *n* (%)	
	ER + Adjuvant	Comparator	ER + Adjuvant	Comparator	ER + Adjuvant	Comparator	ER + Adjuvant	Comparator	ER + Adjuvant	Comparator
ER + adjuvant (>50%) vs. esophagectomy										
Ballard 2016 [22]	T1bSM1 (4), T1b SM2/3 (5)	T1bSM1 (1), T1bSM2/3 (6)	AC (100.0)	AC (100.0)	-	-	Proximal 2/3 (1), Distal 1/3 (8)	Proximal 2/3 (1), Distal 1/3 (6)	1 (11.1)	0 (0.0)
Berger 2023 [23]	M3-SM1 (27), SM2 (19)	M3-SM1 (7), SM2 (8)	SCC (100.0)	SCC (100.0)	25 (5–65)	30 (10–81.7)	-	-	9 (19.6)	2 (13.3)
Emi 2022 [24]	T1a (12), T1bSM1 (9), T1bSM2 (16)	T1a (6), T1bSM1 (7), T1bSM2 (17)	SCC (100.0)	SCC (100.0)	-	-	4/4/20/8/1	1/4/16/9/0	22 (59.5) *	16 (53.3) *
Huang 2022 [25]	T1a (20), T1b (23)	T1a (33), T1b (59)	SCC (100.0)	SCC (100.0)	40 (-)	20 (-)	0/13/27/6/0	0/21/60/11/0	5 (10.9)	9 (9.8)
Ikeda 2015 [16]	T1bSM1 (3), T1bSM2 (4), T1bSM3 (4)	T1bSM1 (7), ≥T1bSM2 (8)	SCC (100.0)	SCC (100.0)	44 (-)	39 (-)	0/0/9/2/0	-	8 (72.7)	6 (40.0)
Kadota 2022 [26]	T1a (10), T1bSM1 (8), T1bSM1 (32)	T1a (2), T1bSM1 (3), T1bSM2 (13)	SCC (100.0)	SCC (100.0)	27 (8–72)	29.5 (12–77)	0/11/30/9 (combined Lt & Ae)	1/1/12/4 (combined Lt & Ae)	29 (58.0)	12 (66.7)
Kanie 2021 [27]	T1a (19), T1bSM1 (13), T1bSM2 (20)	T1a (18), T1bSM1 (6), T1bSM2 (30)	SCC (100.0)	SCC (100.0)	15.0 (4–63)	22.5 (5–74)	0/9/30/13/0	0/15/21/20/0	32 (61.5) *	49 (87.5) *
Katada 2023 [28]	T1aMM with LVI, T1bSM1 NS **, ***	T1aMM with LVI, T1bSM1 NS **, ***	SCC (100.0)	SCC (100.0)	-	-	-	-	-	-
Koterazawa 2018 [29]	T1a (3), T1b (28)	T1a (8), T1b (20)	SCC (100.0)	SCC (100.0)	29 (7–80)	34 (9–100)	2/4/20/5/0	0/3/20/4/1	14 (44.7) *	17 (60.6) *
Li 2023 [30]	T1aMM (3), T1bSM1 (36) **		SCC (100.0)	SCC (100.0)	41 ± 13 **, ****		Lower thoracic 13/Other 26 ***		14 (35.9) **	
Minamide 2023 [32]	T1b (7)	T1b (3)	SCC (100.0)	SCC (100.0)	58 (49–65) ***		0/0/6/4/0 ***		3 (30%) **	
Miyata 2021 [33]	T1a (37), T1bSM1 (19), T1bSM2 (67)	T1a (11), T1bSM1 (7), T1bSM2 (19)	SCC (100.0)	SCC (89.2)	-	-	0/19/69/35/0	0/4/20/13/0	76 (61.8) *	24 (64.9) *
Namikawa 2021 [34]	T1b (9)	T1b (5)	SCC (100.0)	SCC (100.0)	24.5 (3–70) **		2/13/42/9/0 ***		-	-
Ota 2003 [35]	T1bSM1 (14), T1bSM2 (1), T1bSM2 EM+ (3)	T1bSM1 (21), T1bSM2 (49), T1bSM3 (41)	SCC (100.0)	SCC (100.0)	-	-	0/4/10/4/0	-	11 (61)	-
Otaki 2020 [36]	T1b (73)	T1b (68)	AC (100.0)	AC (100.0)	-	-	-	-	20 (29.4)	25 (37.9)
Rodriguez de Santiago 2024 [37]	T1a (12), T1b (25)	-	-	-	-	-	-	-	-	-
Shimada 2022 [38]	T1a/T1bSM1 LVI- (335), T1a and LVI+/T1bSM2 (129), T1bSM1 LVI+ (129) **		SCC (100.0)	SCC (100.0)	-	-	-	-	258 (43.6) **	
Shimizu 2004 [39]	T1a or T1b (16) *****	T1a or T1b (39) *****	SCC (100.0)	SCC (100.0)	24 ± 1.3 ****	28 ± 1.9 ****	0/4/9/3/0	0/8/21/10/0	-	-
Suzuki 2018a [40]	T1a (4), T1b (12)	T1a (9), T1b (7)	SCC (100.0)	SCC (75.0)	-	-	0/3/8/5/0	0/2/8/2/4	14 (87.5) *	14 (87.5) *
Suzuki 2022 [41]	T1a (11), T1b (23)	T1a (8), T1b (18)	SCC (100.0)	SCC (100.0)	-	-	9 (Ce & Ut)/15/10/0	6 (Ce & Ut)/15/4/1	21 (61.8) *	21 (80.8) *
Tanaka 2019 [42]	T1bSM1 (9), T1bSM2 (24)	T1bSM1 (6) T1bSM2 (13)	SCC (100.0)	SCC (100.0)	22 (5–80)	38 (7–88)	1/3/21/8/0	0/3/11/4/0	16 (48.5) *	17 (89.5) *
Yuan 2024 [43]	-	-	SCC (89.0) **		-	-	-	-	-	-
ER + adjuvant (<50%) vs. esophagectomy										
Arima 2007 [44]	T1bSM2 (9), T1bSM3 (6)	T1bSM2 (14), T1bSM3 (46)	SCC (96.0)	SCC (96.0)	38 (7–80)	38 (5–220)	0/0/13/2/0	0/14/26/19/1	-	-
Matsueda 2022 [31]	-	-	SCC (100.0)	SCC (100.0)	-	-	-	-	-	-
Noordzij 2023 [45]										
Qian 2022 [46]	-	T1 (309), T2 (42)	-	-	-	-	-	-	-	-
Wang 2022 [47]	-	M1/M2 (115), MM/SM (87), HGD (98)	SCC (100.0)	SCC (100.0)	-	3.08 ± 1.72 ****	-	25/1/117/5/54	-	3 (1.5)
Zhang 2024 [48]	-	T1aM3 (4), T1b (28)	SCC (100.0)	SCC (100.0)	-	23.5 (9–50)	-	0/7/11/14/0	-	9 (28.1)
Arima 2007 [44]	T1b (6)	T1b (73)	SCC (100.0)	SCC (100.0)	-	20.6 ± 10.4	-	0/15/40/18/0	-	5 (6.8)
**ER + adjuvant vs. CT/CRT/RT alone**										
Kaburagi 2012 [49]	T1aMM (1), T1bSM2 (5), T1bSM1 (1)	T1aMM (1), T1bSM2 (5), T1bSM1 (1)	SCC (100.0)	SCC (100.0)	-	-	1 (Ce and Ut)/4 (Mt)/1 (Mt and Lt)/1(Lt and Ae)	1 (Ce and Ut)/6 (Mt)/1 (Mt and Lt)/1 (Lt)	-	-
Kawaguchi 2015 [50]	T1aM3 (2), T1bSM1 (4), T1bSM2 (10)	T1aM3 (3), T1bSM1 (15), T1bSM2 (13)	SCC (100.0)	SCC (100.0)	-	-	0/1/10/5/0	2/1/18/10/0	-	-
Lyu 2022 [51]	T1a (2), T1b (16)	Tis (3), T1a (7), T1b (10), unknown (17)	SCC (100.0)	SCC (100.0)	20 (10–70)	40 (10–18)	0/2/10/6/0	0/2/25/10/0	-	-
Minashi 2019 [52]	T1a (14), T1b (69)	-	SCC (100.0)	SCC (100.0)	-	-	-	-	-	-
Miyazaki 2022 [53]	T1bSM1 (13), T1bSM2 (50)	T1b (28)	SCC (100.0)	SCC (100.0)	-	-	-	-	-	-
Suzuki 2018b [54]	T1a (8), T1bSM1 (4), T1bSM2 (16), T1bSM3 (1)	T1a (3), T1b (18)	SCC (100.0)	SCC (100.0)	23 (10–100)	50 (10–200)	0/4/13/12/0	0/4/10/7/0	20 (69)	-
Uchinami 2016 [55]	T1a (6), T1b (65) **		SCC (100.0)	SCC (100.0)	40 (10–300) **		0/12/41/18/0 **		-	-
Yoshimizu 2018 [56]	T1bSM1 (8), ≥T1bSM2 (13)	T1b (43)	SCC (100.0)	SCC (100.0)	22 (5–50)	30 (10–80)	1/4/9/6/1	7/8/23/5/0	9 (42.9) *	N/A
ER + adjuvant vs. ER alone										
Chen 2019 [57]	T1 (30), T1a (29), T1b (14), T2 (15)	T1 (109), T1a (444), T1b (88), T2 (30)	AC (64.1)	AC (75.6)	-	-	8 Up/16 Mid/53 Low/11 Other	20 Up/101 Mid/467 Low/83 Other	-	-
Dermine 2020 [58]	T1a (1), T1b (27)	T1a (8), T1b (5)	SCC (92.9)	SCC (76.9)	-	-	0/5/17/6/0	0/0/6/7/0	8 (28.0)	3 (23.0)
Hisano 2018 [59]	T1a (6), T1b-SM1 (7)	T1a (10), T1bSM1 (4)	SCC (100.0)	SCC (100.0)	-	-	0/3/8/2/0	0/2/11/1/0	1 (7.7)	0 (0.0)
Konishi 2024 [15]	-	-	SCC (100.0)	SCC (100.0)	-	-	-	-	-	-
Ren 2022 [60]	T1b (11), T2 (17)	T1b (6), T2 (6), T3 (1)	SCC (100.0)	SCC (100.0)	-	-	-	-	-	-
Suzuki 2021 [61]	T1a (29), T1bSM1 (11), T1bSM2 (24)	T1a (58), T1bSM1 (10), T1bSM2 (14)	SCC (100.0)	SCC (100.0)	32.6 ± 17.8 ****	32.0 ± 17.9 ****	1/5/22/11/0	1/3/22/13/0	13 (33.3)	12 (30.8)
Yang 2023 [62]	T1bSM1 (6), T1bSM2 (41)	T1bSM1 (37), T1bSM2 (77)	SCC (100.0)	SCC (100.0)	-	-	0/10/20/17/0	3/18/48/44/1	17 (36.2)	24 (21.1)
ER + adjuvant alone										
Cho 2024 [63]	T1a (17), T1bSM1 (19), T1bSM2 (42)	N/A	SCC (100.0)	N/A	28 (4–62)	N/A	2/6/43/18/4	N/A	34 (46.6)	N/A
Greenwald 2010 [64]	T1 (2), T1a (24), T1b (10), T2 (10), T3 (2), T4 (1) **	N/A	AC (95.5) **	N/A	40 (10–12) **	N/A	-	N/A	-	N/A
Hamada 2017 [65]	LP (5), MM (18), SM1 (8), SM2+ (35)	N/A	SCC (100.0)	N/A	-	N/A	3/12/35/15/1	N/A	36 (54.5)	N/A
Ikawa 2019 [66]	T1a (32), SM1 (12), SM2 (52)	N/A	SCC (100.0)	N/A	25 (5–75)	N/A	2/14/53/27/0	N/A	66 (68.8)	N/A
Mochizuki 2011 [67]	MM (8), SM1 (4), SM2 (2)	N/A	SCC (100.0)	N/A	25 (10–55) ****	N/A	0/5/8/1/0	N/A	2 (14.3)	N/A
Nemoto 2005 [68]	T1a (11), T1b (19)	N/A	SCC (100.0)	N/A	-	N/A	0/3/18/9/0	N/A	-	N/A
Nichibuchi 2022 [69]	MP (1), T1a (20), T1b (29)	N/A	SCC (98.0)	N/A	30 (5–100)	N/A	5/4/29/12/0	N/A	-	N/A
Yang 2021 [70]	MM (5), SM1 (2), SM2 (24)	N/A	SCC (96.8)	N/A	-	N/A	0/4/17/10/0	N/A	15 (48.4)	N/A

ER + adjuvant vs. esophagectomy studies were divided into two groups denoting whether patients undergoing adjuvant therapy comprised greater or fewer than 50% of the patients undergoing ER. ER = endoscopic resection; CT = chemotherapy; CRT = chemoradiotherapy; N/A = not applicable; NS = not specified; RT = radiotherapy; “-” = not reported. * Lymphatic and venous invasion reported separately and were added together. ** Not reported separately from other groups. *** Among patients undergoing esophagectomy or ER + adjuvant therapy, 172 were T1aMM and 116 were T1bSM. Among this subset, survival rates were analyzed for T1aMM patients with LVI or T1b. Overall, 21.2% of T1aMM patients had LVI. **** Mean and/or standard deviation (median and range not reported). ***** No subdivision between patients with T1a versus T1b stage.

**Table 5 cancers-17-00680-t005:** Treatment characteristics among patients undergoing endoscopic resection with adjuvant therapy in the included studies.

Study	Primary Endoscopic Resection Type (%)	Adjuvant Therapy, *n*	Chemotherapy, *n* (%)	Radiotherapy, *n* (%)	Chemoradiotherapy, *n* (%)	Immunotherapy, *n* (%)	Other Adjuvant Therapy, *n* (%)	Other Adjuvant Therapy Type
	ER + adjuvant (>50%) vs. esophagectomy							
Ballard 2016 [22]	EMR (100)	9	NR	NR	NR	NR	NR	N/A
Berger 2023 [23]	ESD (51.8)	46	3 (6.5)	9 (19.6)	34 (73.9)	-	-	N/A
Emi 2022 [24]	ESD (78.4)	37	37 (100)	-	-	-	-	N/A
Huang 2022 [25]	NS	46	12 (26.1)	34 (73.9)	-	-	-	N/A
Ikeda 2015 [16]	ESD (100)	11	-	-	11 (100)	-	-	N/A
Kadota 2022 [26]	ESD (88.0)	50	-	-	50 (100)	-	-	N/A
Kanie 2021 [27]	ESD (73.1)	52	-	-	52 (100)	-	-	N/A
Katada 2023 [28]	ESD (78.1) *	131	-	-	131 (100)	-	-	N/A
Koterazawa 2018 [29]	ESD (100)	31	-	-	31 (100)	-	-	N/A
Li 2023 [30]	ESD (100)	14	-	-	14 (100)	-	-	N/A
Minamide 2023 [32]	ESD (100)	7	NR	NR	NR	NR	NR	N/A
Miyata 2021 [33]	ESD (69.9) **	123	-	-	123 (100)	-	-	N/A
Namikawa 2021 [34]	ESD (92.4) *	9	-	2 (22.2)	7 (77.8)	-	-	N/A
Ota 2003 [35]	EMR (100)	13	6 (46.2)	2 (15.4)	5 (38.4)	-	-	N/A
Otaki 2020 [36]	EMR (100)	49	-	-	19 (38.8)	-	30 (61.2)	Ablative ***
Rodriguez de Santiago 2024 [37]	ESD (100)	11	1 (9.1)	6 (54.5)	4 (36.4)	-	-	N/A
Shimada 2022 [38]	ESD (100)	155	6 (3.9)	31 (20.0)	117 (75.5)	-	1 (0.6)	Proton beam therapy
Shimizu 2004 [39]	EMR (100)	16	-	-	16 (100)	-	-	N/A
Suzuki 2018a [40]	ESD (100)	16	-	-	16 (100)	-	-	N/A
Suzuki 2022 [41]	ESD (100)	34	-	-	34 (100)	-	-	N/A
Tanaka 2019 [42]	ESD (100)	33	-	-	33 (100)	-	-	N/A
Yuan 2024 [43]	ESD (100)	8	NR	NR	NR	NR	NR	NR
	ER + adjuvant (<50%) vs. esophagectomy							
Arima 2007 [44]	EMR (100)	15	-	-	15 (100)	-	-	N/A
Matsueda 2022 [31]	ESD (77.8) *	91	1 (1.1)	-	90 (98.9)	-	-	N/A
Noordzij 2023 [45]	NS	14	-	-	14 (100)	-	-	N/A
Qian 2022 [46]	ESD (100)	10	1 (10.)	9 (90.0)	-	-	-	N/A
Wang 2022 [47]	ESD (100)	7	-	1 (14.3)	6 (85.7)	-	-	N/A
Zhang 2024 [48]	ESD (100)	6	-	-	6 (100)	-	-	N/A
	**ER + adjuvant vs. CT/CRT/RT alone**							
Kaburagi 2012 [49]	NS	7	-	-	7 (100)	-	-	N/A
Kawaguchi 2015 [50]	ESD (100)	16	-	-	16 (100)	-	-	N/A
Lyu 2022 [51]	NS	18	NR	NR	NR	NR	NR	N/A
Minashi 2019 [52]	ESD (80.1)	83	-	-	83 (100)	-	-	N/A
Miyazaki 2022 [53]	ESD (98.4)	63	-	-	63 (100)	-	-	N/A
Suzuki 2018b [54]	ESD (100)	29	-	1 (3.4)	28 (96.6)	-	-	N/A
Uchinami 2016 [55]	ESD (93.3)	45	-	6 (13.3)	39 (86.7)	-	-	N/A
Yoshimizu 2018 [56]	EMR (61.9)	21	-	-	21 (100)	-	-	N/A
	ER + adjuvant vs. ER alone							
Chen 2019 [57]	NS	88	-	64 (72.7)	24 (27.3)	-	-	N/A
Dermine 2020 [58]	NS	28	-	8 (28.6)	20 (71.4)	-	-	N/A
Hisano 2018 [59]	ESD (100)	17	4 (23.5)	13 (76.5)	-	-	-	N/A
Konishi 2024 [15]	ESD (100)	48	1 (2.1)	7 (14.6)	40 (83.3)	-	-	N/A
Ren 2022 [60]	NS	28	-	-	28 (100)	-	-	N/A
Suzuki 2021 [61]	ESD (100)	39	-	-	39 (100)	-	-	N/A
Yang 2023 [62]	ESD (100)	47	-	47 (100)	-	-	-	N/A
	ER + adjuvant alone							
Cho 2024 [63]	ESD (78.1)	73	-	-	73 (100)	-	-	N/A
Greenwald 2010 [64]	NS	27	-	-	-	-	27 (100)	Cryotherapy
Hamada 2017 [65]	ESD (81.8)	66	-	-	66 (100)	-	-	N/A
Ikawa 2019 [66]	ESD (80.2)	96	-	-	96 (100)	-	-	N/A
Mochizuki 2011 [67]	ESD (100)	14	-	-	14 (100)	-	-	N/A
Nemoto 2005 [68]	EMR (100)	30	-	21 (70.0)	9 (30.0)	-	-	N/A
Nichibuchi 2022 [69]	ESD (100)	50	-	5 (10.)	45 (90.0)	-	-	N/A
Yang 2021 [70]	ESD (100)	31	-	25 (80.6)	6 (19.4)	-	-	N/A

ER + adjuvant vs. esophagectomy studies were divided into two groups denoting whether patients undergoing adjuvant therapy comprised greater or fewer than 50% of the patients undergoing ER. ER = endoscopic resection; EMR = endoscopic mucosal resection; ESD = endoscopic submucosal dissection; N/A = not applicable; NR = not reported; NS = not specified. “-” no patients received designated therapy type. * Not distinguished from other groups. ** While not explicitly reported, 86/123 (69.9%) patients in the endoscopic resection plus adjuvant therapy group were SM1 or SM2. *** Radiofrequency ablation (11), spray cryotherapy (6), photodynamic therapy (4), argon plasma coagulation (4), or combination radiofrequency and thermal (5).

## Data Availability

Data are available on request from the corresponding author.

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
