# Peer review of "Esophagectomy Versus Endoscopic Resection with Adjuvant Therapy for T1b/T2 Esophageal Cancer: A Systematic Review and Meta-Analysis"

_cancers, 2025, doi:10.3390/cancers17040680_

Round 1
Reviewer 1 Report
Comments and Suggestions for Authors
This is a study of contemporary interest to the field and an important question.
The authors have undertaken a thorough review and identified many relevant studies to include in a meta-analysis.
The most significant confounder of case selection, nicely demonstrated as present in all studies in their table 2, has not been adequately addressed.
I think they would be better to focus the question more to try to obtain a more homogeneous patient population.
Minor comments
Introduction.
It is not correct to state that the minority of patients undergoing esophagectomy will survive 3 years or more. Current evidence (FLOT4 trial, ESOPEC among others) suggests in an unstratfied population undergoing neoadjuvant therapy and esophagectomy >50% will survive 5 years. At 3 years this is close to 75% and is higher still for a T1b/T2 only cohort.
“Medical inoperability” is an odd term- rephrase to perhaps medical comorbidity.
Results
Table 1 – needs to include the number of patients in each study.
Major comments
Methodology
“. If studies combined patients undergoing endoscopic resection alone with endoscopic resection plus adjuvant therapy and did not report their outcomes separately, they were included in the quantitative analysis only if patients undergoing adjuvant therapy comprised greater than 50% of the sample”
As the main study aim is to compare patients having endoscopic resection + adjuvant treatment and esophagectomy it is concerning they have included patients having endoscopic resection only (presumably with more favourable pathology) in the comparison.
“As decided a priori, random effects modelling was used unless there was minimal heterogeneity. “
In compiling small clinical observational cohorts for this review from 20 year period across 7 countries there is likely to significant clinical heterogeneity and I doubt a fixed effect model is appropriate at any point.
The majority of studies and included patients are from Japan/China. As reported, the vast majority of patients have squamous cell carcinoma of the esophagus where good outcomes can be achieved with chemoradiation alone and therefore adjuvant treatment following endoscopic resection may be expected to have a good outcome. How do they generalise these data to adenocarcinoma of the esophagus which are much more prevalent in the Dutch and US studies? Perhaps a focus on one histology is better.
As the main study aim is to compare patients having endoscopic resection + adjuvant treatment and esophagectomy it is concerning they have included patients having endoscopic resection only (presumably with more favourable pathology) in the comparison.
Results
Table 4 – very difficult to follow as it is dense with information. There appears to be a natural level of case selection with earlier tumours being more commonly treated with endoscopic therapy +/- adjuvant and more advanced tumours being treated surgically.
This appears to be reflected in the authors assessment in table 2.
There is at least one error in Table 3 – the Chen 2019 did not include outcomes from patients having esophagectomy yet from the table it suggests there were 671 such cases.
An example of the selection bias is illustrated from the Noordzij 2023 study. In this study patients with cT1N0M0 oesophageal cancer either had surgery as their primary treatment modality or endoscopic therapy. This evolved naturally over time as reported in the paper. In the surgery group, despite a clinical stage of cT1N0M0, 16% of patients had pT2 or pT3 tumours compared to 1% of the endoscopic group and 15% of the surgery group had nodal metastases. Treatment selection clearly has an important confounding role here which would directly influence the primary and secondary end points of this study.
The authors have grouped patients by the “majority” treatment which is fundamentally flawed e.g. endoscopic therapy alone, endoscopic therapy and adjuvant all grouped together but classed as endoscopic therapy + adjuvant as >50% had adjuvant. Please report and analyse only according to treatment received.
To assess how realistic the outcome data are, i.e. provide a degree of calibration, it would be helpful to have absolute figures for 5 year overall survival for each treatment type in the meta-analysis group in addition to the hazard ratio for the comparison. Please add.
Figure 4 – the weighting heavily favours the smallest study in this group which appears odd but may reflect poor quality data from the larger studies? Please comment.
I think overall the question is good and the answer of interest to the field. It would be preferable if the authors focussed on studies with a clearer baseline cohort. E.g. only patients fit enough for radical therapy having endoscopic treatment with a pathological diagnosis of pT1b esophageal cancer. What is the outcome in overall and disease specific survival for patients treated with observation, adjuvant chemoradiotherapy or then proceeding to esophagectomy?
One problem with their current analysis is that in at least some of the studies, patient undergoing esophagectomy have only had clinical staging and a significant proportion are understaged (as demonstrated above for the Noordzij study). Comparing survival outcomes here is flawed.
A further issue is the cohort of patients who are not fit enough for esophagectomy. The morbidity and outcomes from esophagectomy are not relevant here. The comparison should be between endoscopic treatment alone and endoscopic treatment followed by adjuvant treatment.
I hope they can revise this work to provide a solid baseline to then support a randomised trial in this area as I suspect this is needed to overcome the treatment selection bias.
Author Response
Comment 1:
This is a study of contemporary interest to the field and an important question.
The authors have undertaken a thorough review and identified many relevant studies to include in a meta-analysis.
The most significant confounder of case selection, nicely demonstrated as present in all studies in their table 2, has not been adequately addressed.
I think they would be better to focus the question more to try to obtain a more homogeneous patient population.
Response 1:
Thank you for these important points. We totally agree with the reviewer about the significant residual confounding. However, we believe that the data and practice clearly show that the selection bias is still strongly favoring esophagectomy (as those who are treated with ESD/EMR + adjuvant therapy are clearly not fit enough to tolerate or be offered esophagectomy). Thus, a finding of no significant differences serves to highlight that there is truly equipoise in this area and warrants randomized trials. We believe that including the entire population helps to make that point. For that reason, we advocate to keep all the population.
Comment 2:
It is not correct to state that the minority of patients undergoing esophagectomy will survive 3 years or more. Current evidence (FLOT4 trial, ESOPEC among others) suggests in an unstratfied population undergoing neoadjuvant therapy and esophagectomy >50% will survive 5 years. At 3 years this is close to 75% and is higher still for a T1b/T2 only cohort.
Response 2:
We thank the reviewer for their feedback and insight on our manuscript. At the time of our writing our paper, FLOT4 and ESOPEC had not been published. We agree with the reviewer’s assessment. We have removed this sentence and the corresponding reference from lines 59-61.
Comment 3:
“Medical inoperability” is an odd term- rephrase to perhaps medical comorbidity.
Response 3:
We thank the reviewer for identifying this point. “Inoperability” has been replaced with “comorbidities rendering patient inoperable” (line 73).
Comment 4:
Table 1 – needs to include the number of patients in each study.
Response 4:
We thank the reviewer for raising this point. This information is contained in Table 3, which includes the number of eligible patients from each study in its second column. As the sample sizes are pertinent to contextualizing other demographic information in Table 3 such as age and sex, we respectfully believe that the number of patients in each study is more suitably included in Table 3. If the reviewer or editors feel strongly that it should be moved to Table 1, please inform us and we would be happy to reassess this point.
Comment 5:
“. If studies combined patients undergoing endoscopic resection alone with endoscopic resection plus adjuvant therapy and did not report their outcomes separately, they were included in the quantitative analysis only if patients undergoing adjuvant therapy comprised greater than 50% of the sample
As the main study aim is to compare patients having endoscopic resection + adjuvant treatment and esophagectomy it is concerning they have included patients having endoscopic resection only (presumably with more favourable pathology) in the comparison.
Response 5:
We thank the reviewer for their feedback. This is an important point that we did not satisfactorily clarify in our manuscript. Reassuringly, 21/23 of the studies in the >50% group separately identified outcomes of ER + adjuvant patients from ER alone patients (i.e. constituting 100% ER + adjuvant). Of the 2/23 that did not separate their outcomes reporting, 72.2% (Ota, 2003) and 67.1% (Otaki 2020) were ER + adjuvant. Nevertheless, this was not distinguished properly our manuscript. As such, we have added a reference to this limitation in lines 470-77.
Furthermore, in this situation, the bias exists against ER + adjuvant and thus the meta-analysis results showing no significant difference compared to esophagectomy lend further credibility to the idea that there is clinical equipoise for RCT.
Comment 6:
“As decided a priori, random effects modelling was used unless there was minimal heterogeneity. “
In compiling small clinical observational cohorts for this review from 20 year period across 7 countries there is likely to significant clinical heterogeneity and I doubt a fixed effect model is appropriate at any point.
Response 6:
We thank the reviewer for this point and agree with their assessment. We only did this as this was pre-determined a priori by our protocol. We have amended the modelling for all analyses to be random effects. This affects Figures 4 and 7, which were previously fixed effects and are now random effects. This does not affect HR, CI, or p-value or Figure 4, but does affect those of Figure 7, which have been denoted in red. It does not change the overall determination of significance for either analysis (i.e. both remain nonsignificant). The reference to the a priori decision to use fixed effects in event of minimal heterogeneity has been removed from the Methods section (lines 151-152) and has been replaced with statement “Random effects modelling was used.”
Comment 7:
The majority of studies and included patients are from Japan/China. As reported, the vast majority of patients have squamous cell carcinoma of the esophagus where good outcomes can be achieved with chemoradiation alone and therefore adjuvant treatment following endoscopic resection may be expected to have a good outcome. How do they generalise these data to adenocarcinoma of the esophagus which are much more prevalent in the Dutch and US studies? Perhaps a focus on one histology is better.
Response 7:
We thank the reviewer for raising this important point. We agree that SCC and AC cannot be put forth as if the same. As we described in our study protocol, it was our hope to perform subgroup analysis according to histology. Unfortunately, only 2 of the studies included in the meta-analysis described patients with AC and only 1 of which reported on any of our primary outcomes. As such, subgroup analysis according to histology was not feasible. Reassuringly though, this means that we have near-homogeneity among the studies included in the meta-analysis. Furthermore, this analysis is more reflective of the disproportionate burden of esophageal disease, as East Asian populations are at greater risk for esophageal cancer and disproportionately suffer SCC as opposed to AC. We would refer to the following study for further context:
Huang J, Koulaouzidis A, Marlicz W, et al. Global Burden, Risk Factors, and Trends of Esophageal Cancer: An Analysis of Cancer Registries from 48 Countries. Cancers (Basel). 2021;13(1):141. Published 2021 Jan 5. doi:10.3390/cancers13010141.
While it would be preferable to focus on this one histology, we would conversely risk overlooking the full burden of esophageal cancer worldwide. This is especially considering we are bereft of data from which to perform analysis from the outset. As such, it may be more prudent to repeat a systematic review of this nature at a future date once there are more studies published that compare ER + adjuvant therapy to esophagectomy in both AC and SCC. This would fully capture the differences in outcomes based on histology. However, outcomes reporting stratified according to histology is unfortunately beyond the scope of this systematic at present
Overall, we agree with the reviewer regarding this crucial point. As such, we have added a reference to this in our Limitations section on lines 464-467 and to our Conclusions section on lines 511-512.
Comment 8:
As the main study aim is to compare patients having endoscopic resection + adjuvant treatment and esophagectomy it is concerning they have included patients having endoscopic resection only (presumably with more favourable pathology) in the comparison.
Response 8:
We thank the reviewer for raising this important point. We regret we were vague in this regard. To clarify, we included studies with patients undergoing endoscopic resection alone as comparison group as these studies nevertheless did fulfill our inclusion criteria on the basis of intervention group (i.e. ER + adjuvant therapy). We did not exclude studies on the basis of comparison group (i.e. esophagectomy, ER alone, nothing). This was so as to maximize the sensitivity of our search in what was anticipated to be an area with relatively few samples available from which to draw.
As such, while Tables 2-5 do report comparison groups that are ER alone, none of these studies are included in the meta-analysis. They are for descriptive purposes only and for transparent reporting from what we found in the search that met inclusion criteria. Only studies where ER + adjuvant could be compared to esophagectomy were included in the meta-analysis.
While it would have been ideal to ultimately exclude studies that did not directly compare to esophagectomy after the fact, this would have been a post hoc addendum to our a priori study protocol. Furthermore, we concluded it was also perhaps useful to retain these additional studies simply for the descriptive purposes outline in Tables 2-5 and compare/contrast them to the studies that compared ER + adjuvant to esophagectomy.
Overall, we agree with the reviewer. As such, we have summarized the descriptive purpose of studies not comparing to esophagectomy and clarified their exclusion from meta-analysis in Lines 138-141.
Comment 9:
Table 4 – very difficult to follow as it is dense with information. There appears to be a natural level of case selection with earlier tumours being more commonly treated with endoscopic therapy +/- adjuvant and more advanced tumours being treated surgically.
This appears to be reflected in the authors assessment in table 2.
Response 9:
We thank the reviewer for this point. We agree that there could be a natural level of case selection at play by virtue of the observational nature of the included studies. Fortunately, on review of the reported TNM staging of patients treated with ER + adjuvant versus esophagectomy in Table 4, there appear to be comparable proportions of disease severity (i.e. the majority in both groups are T1b). If we have overlooked a particular subset, please inform us and we would be eager to reassess this. The risk of bias due to confounding and selection of participants as identified in Table 2 is certainly serious, and is also multifactorial due to the observational, and predominantly retrospective, nature of all of the included studies. Overall, this is worth elaborating on further, and we have added this description to Limitations lines 458-462.
Regarding the density of Table 4, we agree with the reviewer. That said, we elected to retain it as opposed to splitting the table into multiple subdivisions because this would further increase the table burden on the manuscript (i.e. would either have to repeat the same 58 rows again if subdivided according to column or would have to repeat the same 11 columns again if subdivided according to row). Overall, we must report on the tumour characteristics in some way. Unfortunately, this appears to be the most parsimonious option we have. If there are recommendations to optimize Table 4, we would be eager to implement any feedback. Alternatively, we would be happy to submit the table in whole/part as an appendix if preferred by the editorial team.
As such, we will respectfully elect to defer changes to Table 4 at this time. However, if formatting changes are required by the editorial staff, please let us know and we would be happy to reassess according to the preference that would best suit the desired formatting.
Comment 10:
There is at least one error in Table 3 – the Chen 2019 did not include outcomes from patients having esophagectomy yet from the table it suggests there were 671 such cases.
Response 10:
We thank the reviewer for raising this point. This is our error in how we have presented the table as opposed to the data abstraction itself. The column title the reviewer refers to is “Esophagectomy” but the patients in the study are under the “ER + adjuvant vs ER alone” subheading. 671 of patients in the Chen 2019 study underwent ER alone (as did 13 in Dermine 2020, 14 in Hisano 2018, etc.).
The column title should more accurately state “Comparator” (with esophagectomy, CRT, ER alone, or none stated in the subheading).
As such, we have changed all “Esophagectomy” column titles to “Comparator” in Tables 3 and 4 to more accurately denote this while maintaining the subheadings that define the comparator group.
Comment 11:
An example of the selection bias is illustrated from the Noordzij 2023 study. In this study patients with cT1N0M0 oesophageal cancer either had surgery as their primary treatment modality or endoscopic therapy. This evolved naturally over time as reported in the paper. In the surgery group, despite a clinical stage of cT1N0M0, 16% of patients had pT2 or pT3 tumours compared to 1% of the endoscopic group and 15% of the surgery group had nodal metastases. Treatment selection clearly has an important confounding role here which would directly influence the primary and secondary end points of this study.
Response 11:
We thank the reviewer for their feedback. We agree with the point raised regarding a concern for selection bias. <50% of patients undergoing ER in this study received adjuvant therapy. As such, the Noordzij 2023 study was not included in the meta-analysis. Therefore, it did not play a role in the primary or secondary outcomes in our study. T3 disease was excluded from our analysis as well. Furthermore, selection bias was fortunately ameliorated to a degree as the majority of studies that were included in the meta-analysis reported pathologic stage as opposed to the clinical stage.
Comment 12:
The authors have grouped patients by the “majority” treatment which is fundamentally flawed e.g. endoscopic therapy alone, endoscopic therapy and adjuvant all grouped together but classed as endoscopic therapy + adjuvant as >50% had adjuvant. Please report and analyse only according to treatment received.
Response 12:
We thank the reviewer for this point. We agree that reporting and analyzing according only to the treatment received is preferable. Indeed, the reviewer’s request is already satisfied in 21/23 ER + adjuvant >50% studies as 100% patients received ER + adjuvant therapy in those. We refer to Comment/Response 5 for our prior discussion of this. In brief, in the two studies that did not separate their outcomes reporting from ER versus ER + adjuvant therapy, 72.2% (Ota, 2003) and 67.1% (Otaki 2020) were ER + adjuvant. As such, we deemed that those studies still warranted inclusion instead of rejecting both studies outright. That said, we acknowledge this limitation as before and reference the addition to the Limitations section describing this in lines 470-477.
Comment 13:
To assess how realistic the outcome data are, i.e. provide a degree of calibration, it would be helpful to have absolute figures for 5 year overall survival for each treatment type in the meta-analysis group in addition to the hazard ratio for the comparison. Please add.
Response 13:
We thank the reviewer for this point. We have added the absolute figures for 5-year overall survival in addition to the HR in Figures 2a and 2b (see column 5yOS (%)).
Comment 14:
Figure 4 – the weighting heavily favours the smallest study in this group which appears odd but may reflect poor quality data from the larger studies? Please comment.
Response 14:
Our thanks to the reviewer for their feedback. This is a strong point of interest. The weight (size) of the boxes on a forest plot is derived from sample size, but also from confidence intervals. The smaller the CI and the more consistent the estimate of effect, the more weight a study carries.
While Li 2023 contains the smallest sample size, it’s CI is also the smallest among the other studies (1.09-63.69) as well as being clearly on one side of the null effect line. This is why the relative weights for Li 2023 and Katada 2023 are 47.7% and 34.7% respectively, in addition to what is taken into account by their sample sizes.
Comment 15:
I think overall the question is good and the answer of interest to the field. It would be preferable if the authors focussed on studies with a clearer baseline cohort. E.g. only patients fit enough for radical therapy having endoscopic treatment with a pathological diagnosis of pT1b esophageal cancer. What is the outcome in overall and disease specific survival for patients treated with observation, adjuvant chemoradiotherapy or then proceeding to esophagectomy?
Response 15:
We thank the reviewer for their feedback. While it would be prudent to assess our patients in this manner (with separate observation, adjuvant CRT, and esophagectomy groups) or with only patients fit for radical therapy who then undergo endoscopic treatment, unfortunately cohorts are not often reported this way in the literature. This is why we constructed our inclusion criteria the way we did: so as to accommodate for the real-world treatment pathways and reporting that are known for our patients.
The reality is that there are very few studies reporting on short term and even fewer reporting on long term outcomes of patients who are otherwise operable but elect to have ER + adjuvant therapy rather than esophagectomy. Furthermore, none of these studies report on an esophagectomy comparator group. We totally agree with the reviewer about the significant residual confounding. However, we believe that the data and practice clearly show that the selection bias is still strongly favoring esophagectomy (as those who are treated with ESD/EMR + adjuvant therapy are clearly not fit enough to tolerate or be offered esophagectomy). Thus, a finding of no significant differences in outcomes serves to highlight that there is truly equipoise in this area and warrants randomized trials. We believe this will create the conditions to be able to generate data to better answer the question, as the reviewer states.
As ER + adjuvant therapy for T1b cancers becomes more commonplace, we suspect there will be more comparisons available between observation, adjuvant CRT, and esophagectomy. We also suspect that the decision-making between esophagectomy versus ER + adjuvant therapy may become more frequently offered to patients fit for radical therapy. It would be at that point where a revised systematic review would be feasible for this purpose.
However, at present, it is not feasible to reconstruct and reperform the systematic review with these new comparison groups. Currently, to our knowledge, the literature does not report baseline risk and outcomes with this kind of granularity to make it achievable. As well, this type of revision at this stage in a systematic review would also unfortunately violate our a priori prospectively registered study protocol.
As such, we will respectfully defer any changes to the baseline cohort at this time. However, we are eager to repeat this review on a larger scale with more granular baseline cohorts in the future when there is more data to support this.
Comment 16:
One problem with their current analysis is that in at least some of the studies, patient undergoing esophagectomy have only had clinical staging and a significant proportion are understaged (as demonstrated above for the Noordzij study). Comparing survival outcomes here is flawed.
Response 16:
We thank the reviewer for their feedback. We refer to Comment/Response 11 regarding the concern for the Noordzij study. Furthermore, the meta-analysis included groups of patients undergoing esophagectomy where the majority of whom had confirmed pT1b or pT2 disease. If groups were reported to have pT3 or higher disease, they were not included in the analysis. That said, we appreciate the concern regarding possible understaging in general for patients who undergo esophagectomy in terms of real-world applicability. As such, we have added this as a limitation to lines 478-481.
Comment 17:
A further issue is the cohort of patients who are not fit enough for esophagectomy. The morbidity and outcomes from esophagectomy are not relevant here. The comparison should be between endoscopic treatment alone and endoscopic treatment followed by adjuvant treatment.
Response 17:
Our thanks to the reviewer for this feedback. This is another strong point. We totally agree with the reviewer about this point. However, we believe that the data and practice clearly show that the selection bias is still strongly favoring esophagectomy (as those who are treated with ESD/EMR +adjuvant therapy are clearly not fit enough to tolerate or be offered esophagectomy). Thus, a finding of no significant differences in outcomes serves to highlight that there is truly equipoise in this area and warrants randomized trials.
That is the whole point of this type of study, to understand, using all the available data (albeit nonrandomized and flawed by its nature), what is the state of the evidence and to either demonstrate that there is equipoise enough to warrant randomized trials or to identify such a bad signal of harm that it cautions our community away from a randomized trial. As more studies are published, we anticipate that this may become more commonplace as pT1b/T2 is increasingly treated with ER + adjuvant therapy. However, we lack reporting of robust data to substantiate an analysis for this separate subgroup of patients. Again, this would be an excellent comparator group to include in a revised systematic review when there are more studies available (similar to Comment/Response 15). Moreover, revising the review at this point for subgroup analysis on patients who are deemed fit/unfit for surgery is unfortunately beyond the scope of this systematic review at this time. As well, like Comment/Response 15, it would violate our a priori defined prospectively-registered systematic review protocol.
Reviewer 2 Report
Comments and Suggestions for Authors
The authors present a manuscript with pertinent research question, thorough methodology, and clearly presented results, for which they are to be commended.
Despite the good work, I still have a major issue, which is rooted in the data, but not in the authors work. As the authors note (lines 158-160), there is a preponderance of studies from the east. It is well known that esophageal cancers differ between patients from the east and those from the west. The major difference being the histological type of tumor, which is adenocarcinoma in the west. The available evidence suggests that adenocarcinoma is associated with an increased risk for distant metastases that may impact the results. Although it is not investigated, even if it would have been, the data might be unable to demonstrate that difference due to the predominance of studies from the east. For that reason I don't feel that another post-hoc analysis would be suitable to address that issue. Given this limitation - which is not discussed - the conclusions seem to be a bit overambitious. The data is the limitation precluding the statement in its current form and thus requires reformulation that this finding might not be applicable to adenocarcinoma given the high prevalence of squamous cell carcinoma in the data the authors included, because this fact limits the generalizability of the results.
Author Response
Comment 1:
The authors present a manuscript with pertinent research question, thorough methodology, and clearly presented results, for which they are to be commended.
Despite the good work, I still have a major issue, which is rooted in the data, but not in the authors work. As the authors note (lines 158-160), there is a preponderance of studies from the east. It is well known that esophageal cancers differ between patients from the east and those from the west. The major difference being the histological type of tumor, which is adenocarcinoma in the west. The available evidence suggests that adenocarcinoma is associated with an increased risk for distant metastases that may impact the results. Although it is not investigated, even if it would have been, the data might be unable to demonstrate that difference due to the predominance of studies from the east. For that reason I don't feel that another post-hoc analysis would be suitable to address that issue. Given this limitation - which is not discussed - the conclusions seem to be a bit overambitious. The data is the limitation precluding the statement in its current form and thus requires reformulation that this finding might not be applicable to adenocarcinoma given the high prevalence of squamous cell carcinoma in the data the authors included, because this fact limits the generalizability of the results.
Response 1:
We thank the reviewer for their feedback and insight on our manuscript. This comment is in line with Comment 7 from Reviewer 1.
We agree that SCC and AC cannot be put forth as if the same. It was our intent to perform subgroup analysis according to histology (please refer to our a priori published protocol). Unfortunately, only 2 of the studies included in the meta-analysis described patients with AC and only 1 of which reported on any of our primary outcomes. As such, subgroup analysis according to histology was not feasible. Reassuringly though, this means that we have near-homogeneity among the studies included in the meta-analysis.
While it would be preferable to focus on one histology (i.e. SCC), we would also risk overlooking the full burden of esophageal cancer worldwide among the few studies that do report on AC. This is especially considering we are bereft of data from which to perform this analysis from the outset. As such, it may be more prudent to repeat a systematic review of this nature at a future date once there are more studies published that compare ER + adjuvant therapy to esophagectomy in both AC and SCC. This would fully capture the differences in outcomes based on histology. However, outcomes reporting stratified according to histology is unfortunately beyond the scope of this systematic review at present
Overall, we agree with the reviewer regarding this crucial point. As such, we have added a reference to this in our Limitations section on lines 464-467 and to our Conclusions in lines 511-512.
Reviewer 3 Report
Comments and Suggestions for Authors
The manuscript addresses a critical clinical question, exploring the potential for endoscopic resection with adjuvant therapy as an alternative to esophagectomy for T1b/T2 esophageal cancer. This topic is both timely and relevant, given the significant morbidity associated with esophagectomy. The systematic approach, adherence to PRISMA guidelines, and PROSPERO registration strengthen the study's methodological rigor. Additionally, the inclusion of subgroup analyses for survival and recurrence outcomes offers valuable insights into treatment efficacy for different patient populations.
However, some areas need further clarification to enhance the manuscript's impact. First, the rationale for excluding studies with fewer than five patients undergoing endoscopic resection and adjuvant therapy is unclear. Providing a justification for this cutoff would improve the transparency of the selection criteria. Additionally, while the risk-of-bias assessment is comprehensive, the high prevalence of serious risk of bias in the included studies should be discussed more thoroughly in the limitations section to contextualize the findings. Finally, further elaboration on the clinical implications of subgroup analyses—particularly regarding lymphovascular invasion—would strengthen the discussion and help guide future clinical practice. Addressing these points will enhance the robustness and applicability of the study's conclusions.
Author Response
Comment 1:
The manuscript addresses a critical clinical question, exploring the potential for endoscopic resection with adjuvant therapy as an alternative to esophagectomy for T1b/T2 esophageal cancer. This topic is both timely and relevant, given the significant morbidity associated with esophagectomy. The systematic approach, adherence to PRISMA guidelines, and PROSPERO registration strengthen the study's methodological rigor. Additionally, the inclusion of subgroup analyses for survival and recurrence outcomes offers valuable insights into treatment efficacy for different patient populations.
However, some areas need further clarification to enhance the manuscript's impact.
First, the rationale for excluding studies with fewer than five patients undergoing endoscopic resection and adjuvant therapy is unclear. Providing a justification for this cutoff would improve the transparency of the selection criteria.
Response 1:
We thank the reviewer for their feedback and insight on our manuscript. We agree. The rationale for excluding studies with <5 patients undergoing ER + adjuvant therapy was arbitrarily defined. The intent was to reduce the likelihood of including case series that were otherwise categorized as cohort studies, while also permitting for cohort studies with low sample sizes but valid calculations of survival and risk to be included (e.g. Li 2023 as included in our meta-analysis).
We refer to the following review that encapsulates this grey area between case series and cohort studies more thoroughly:
Dekkers OM, Egger M, Altman DG, Vandenbroucke JP. Distinguishing case series from cohort studies. Ann Intern Med. 2012;156(1 Pt 1):37-40. doi:10.7326/0003-4819-156-1-201201030-00006.
Overall, there is no clear consensus for the boundary between case series and cohort studies that is widely agreed-upon. As such, we implemented this arbitrary threshold to minimize likelihood of including inappropriate case series, while also permitting valid low-sample cohort studies to be included.
We have added a justification for this cutoff to lines 109-113.
Comment 2:
Additionally, while the risk-of-bias assessment is comprehensive, the high prevalence of serious risk of bias in the included studies should be discussed more thoroughly in the limitations section to contextualize the findings.
Response 2:
We thank the reviewer for raising this point. We agree. In addition to the reference to limited interpretability due to the cohort study-predominance in lines 457-458, we have also expanded on this in lines 458-462.
Comment 3:
Finally, further elaboration on the clinical implications of subgroup analyses—particularly regarding lymphovascular invasion—would strengthen the discussion and help guide future clinical practice. Addressing these points will enhance the robustness and applicability of the study's conclusions.
Response 3:
We thank the reviewer for their feedback and agree with their assessment. In addition to our discussion of the lymphovascular invasion subgroup in lines 383-387 and 419-426, we have added the further clinical implications of this finding to lines 426-432. Unfortunately, we do not feel like we can rightfully make conclusions on relevance to future clinical practice beyond what we have written as our review remains subverted by the high risk of bias, disproportion of SCC to AC pathology, and the need for RCT-level evidence to substantiate the findings
Round 2
Reviewer 1 Report
Comments and Suggestions for Authors
By and large the major comments have not been addressed to revise the results and conclusions but the authors have acknowledged these and added some additional limitations to their discussion.
Minor comments
FLOT4 trial and survival outcome has been reported for some time.
Inoperability relates to tumour factors e.g. T4b disease. Comorbidity relates to risk of complications and death. The introduction sentence still does not make sense.
The survival plots have not come through in the revised manuscript.
Author Response
Thank you for your consideration of our manuscript. We are certainly committed to doing the best we can to address your concerns but there is a limit to the existing data. Furthermore, we are able to do a lot of things analytically but fundamentally there will always be residual selection bias and the confounding by indication in the existing literature. By its nature, the decision to offer endoscopic resection and adjuvant therapy rather than the standard of care esophagectomy can be assumed to be a biased decision that occurs the patient is considered too high risk and unlikely to do well with esophagectomy; thus, any study other than a randomized trial cannot eliminate the fundamental confounding by indication. Our main purpose of performing and publishing this study is to demonstrate this fact and to demonstrate that despite the fact that the residual bias should favour esophagectomy, we are seeing a signal of similar outcomes; this means we should be considering randomized studies to better answer the question. We need studies like ours to provide an empiric demonstration and justification for this; thus, we expect this will be a highly cited study as more groups start addressing this question.
Comment 1:
By and large the major comments have not been addressed to revise the results and conclusions but the authors have acknowledged these and added some additional limitations to their discussion.
Response 1:
We thank the reviewer for their feedback and agree with their assessment. While we are uncertain which of the major comments from Reviewer 1 (i.e. Comments 5-17 from Round 1) or their parts were not adequately addressed, we will endeavour to improve our revisions herein.
A caveat: we are not able to necessarily change the Results at will. This is because these are values obtained from data extracted and analyzed from previous studies based on an a priori registered analysis protocol that has already been completed. To modify the Results based on what we would prefer to present after the fact could be considered tantamount to subverting the systematic review protocol registration process itself (thus violating the PRISMA guidelines and the point of prospective registration on the PROSPERO database to begin with).
However, we have made the following modifications to the Results, Discussion, and Conclusions section to reflect the points raised in the major comments for their revision:
- Results: Regarding the concern around SCC versus AC in Round 1 Comment 7 by Reviewer 1 and Comment 1 by Reviewer 2:
- As before, only 2 of the studies included in the meta-analysis described patients with AC with only 1 of which that reported on any of our primary outcomes. That said, we understand that this was a concern for multiple reviewers.
- As such, we repeated the analysis for all outcomes based on subgroups of SCC versus AC and have included this as a new Supplementary Appendix D Figures 1-4 and a description in Results Lines 421-431.
- The 5-year OS and DFS favours esophagectomy for the single AC subgroup study (Otaki 2020) and favours ER + adjuvant therapy for the SCC subgroup. CSS was unable to be analyzed as all studies that reported 5-year CSS were SCC. However, the major caveat to this is that the adjuvant therapy in this AC study was predominantly RFA, which has in multiple subgroups been identified as conferring lower survival and theoretically also can be understood to confer poor locoregional control as opposed to adjuvant or chemotherapy. Thus, this AC subgroup is highly unreliable.
- No differences in recurrence or perioperative mortality according to subgroup. Unable to assess adverse events because no AC studies reported on these.
- While these 5yOS and DFS results are intriguing and should be explored in future studies once more data from adenocarcinoma cohorts are available, we cannot draw any firm conclusions that esophagectomy is favoured for AC patients on the basis of one study.
- Conversely, whether the AC study is grouped in with the SCC studies, it does not change the results of the overall analysis for the SCC patients (i.e. no difference in 5-year OS, DFS, recurrence, or perioperative mortality regardless of whether AC study is included).
- Therefore, this post-hoc analysis does not materially change conclusions for the outcomes of interest.
- Certainly, this may be of interest to note and we have added it in the Discussion Lines 519-524.
- Discussion:
- We have added to the first paragraph of Discussion that our results are limited in their interpretability due to the high risk of bias and heterogeneity within the studies (Lines 448-450).
- We have described the implications of heterogeneity of studies on clinical applicability and future directions in Lines 519-524.
- We highlight the existing descriptions around heterogeneity in histology, outcomes reporting, and treatment selection as limitations to the interpretation of our results in Lines 533-546.
- Conclusion:
- Added sentence to Conclusion identifying limited clinical applicability due to heterogeneity (Lines 592-594).
While not a primary consideration for our revisions, in the interest of the word count, we will hold off on further additions to our Results, Discussion, and Conclusions. While there are several more areas we could expound, we note that Discussion and Conclusion alone have now exceeded 1700 words. If there are specific points that the reviewers/editorial staff would like us to elaborate on further, we would be happy to do so.
Comment 2:
Minor comments
FLOT4 trial and survival outcome has been reported for some time.
Response 1:
We thank the reviewer for their feedback. We agree with their assessment of the survival outcomes from those studies and acknowledge their publication dates. We note the point of misunderstanding was on our part due to our unclear description of the health-related quality of life outcome on the basis of patients surviving 3 or more years postoperatively from the following study:
Katz A, Nevo Y, Ramírez García Luna JL, et al. Long-Term Quality of Life After Esophagectomy for Esophageal Cancer. Ann Thorac Surg. 2023;115(1):200-208. doi:10.1016/j.athoracsur.2022.07.029
Our intent was not to comment on survival by citing this paper, but to comment on health-related quality of life (which happened to be described among surviving patients in this prospective database, of whom 47% survived 3 or more years).
As such, our previous statement describing the results of this study will remain omitted in order to prevent inadvertently misleading readership.
Comment 2:
Inoperability relates to tumour factors e.g. T4b disease. Comorbidity relates to risk of complications and death. The introduction sentence still does not make sense.
Response 2:
We thank the reviewer for their feedback.
We removed the previous term “inoperability” and attempted to clarify “medical comorbidities rendering patient’s inoperable” in Line 73 based on Comment 2 from Reviewer 1 in Round 1, stating:
“‘Medical inoperability’ is an odd term- rephrase to perhaps medical comorbidity.”
However, we note and agree with the reviewer’s kind distinction between inoperability and comorbidity.
We will therefore omit the phrase “rendering patient’s inoperable” so that it is in line with the original suggestion made by Reviewer 1 from Round 1.
To ensure we do not misunderstand, we would please ask the reviewers/editors to confirm if “introduction sentence” here is referring instead to Lines 54 or 68 so we may change those if they are the sentences with which there remain concerns.
Comment 3:
The survival plots have not come through in the revised manuscript.
Response 3:
We thank the reviewer for their feedback. As far as we can tell, there was not a comment regarding survival plots from the previous round of review. The most similar point we can ascertain is from Comment 11 from Reviewer 1 that read:
“To assess how realistic the outcome data are, i.e. provide a degree of calibration, it would be helpful to have absolute figures for 5 year overall survival for each treatment type in the meta-analysis group in addition to the hazard ratio for the comparison. Please add.”
To this end, we did add the absolute values for 5-year overall survival for each treatment type in the meta-analysis in addition to the hazard ratio for the comparison. These values can be found in the fourth and sixth columns from the left of both figures that describe 5-year overall survival (labelled “5yOS (%) in Figures 2a and 2b).
If the feedback is instead to provide new diagrams that display the 5-year overall survival curves for each study included in the meta-analysis, we unfortunately do not think this would be feasible. This is because (1) It would add at least 10 new figures to the manuscript, all of which would potentially be subject to copyright from their respective journals even if reproduced by ourselves due to their likeness and (2) Would not display the absolute 5-year overall survival values regardless (as survival curves from Kaplan-Meier diagrams do not portray absolute values of survival to begin with, but are rather estimated survival probabilities).
If we misunderstand the reviewer’s comment, please let us know and we would be happy to reassess.